# *ahctf1* and *kras* mutations combine to amplify oncogenic stress and restrict liver overgrowth in a zebrafish model of hepatocellular carcinoma

Kimberly J Morgan[1,2], Karen Doggett[1,2], Fansuo Geng[1,2], Stephen Mieruszynski[1,2], Lachlan Whitehead[2,3], Kelly A Smith[4,5], Benjamin M Hogan[5,6], Cas Simons[5,7], Gregory J Baillie[5], Ramyar Molania[2,8], Anthony T Papenfuss[2,8], Thomas E Hall[5], Elke A Ober[9], Didier YR Stainier[10], Zhiyuan Gong[11], Joan K Heath[1,2]*

[1]Epigenetics and Development Division, Walter and Eliza Hall Institute of Medical Research, Parkville, Australia; [2]Department of Medical Biology, University of Melbourne, Parkville, Australia; [3]Centre for Dynamic Imaging, Advanced Technology and Biology Division, Walter and Eliza Hall Institute of Medical Research, Parkville, Australia; [4]Department of Physiology, University of Melbourne, Parkville, Australia; [5]Institute for Molecular Biosciences, University of Queensland, Queensland, Australia; [6]Peter MacCallum Cancer Centre, Melbourne, Australia; [7]Murdoch Children's Research Institute, Parkville, Australia; [8]Bioinformatics Division, Walter and Eliza Hall Institute of Medical Research, Parkville, Australia; [9]Danish Stem Cell Center, University of Copenhagen, Copenhagen, Denmark; [10]Department of Developmental Genetics, Max Planck Institute for Heart and Lung Research, Bad Nauheim, Germany; [11]Department of Biological Science, National University of Singapore, Singapore, Singapore

*For correspondence:
joan.heath@wehi.edu.au

**Abstract** The nucleoporin (NUP) ELYS, encoded by *AHCTF1*, is a large multifunctional protein with essential roles in nuclear pore assembly and mitosis. Using both larval and adult zebrafish models of hepatocellular carcinoma (HCC), in which the expression of an inducible mutant *kras* transgene (*kras*$^{G12V}$) drives hepatocyte-specific hyperplasia and liver enlargement, we show that reducing *ahctf1* gene dosage by 50% markedly decreases liver volume, while non-hyperplastic tissues are unaffected. We demonstrate that in the context of cancer, *ahctf1* heterozygosity impairs nuclear pore formation, mitotic spindle assembly, and chromosome segregation, leading to DNA damage and activation of a Tp53-dependent transcriptional programme that induces cell death and cell cycle arrest. Heterozygous expression of both *ahctf1* and *ranbp2* (encoding a second nucleoporin), or treatment of heterozygous *ahctf1* larvae with the nucleocytoplasmic transport inhibitor, Selinexor, completely blocks *kras*$^{G12V}$-driven hepatocyte hyperplasia. Gene expression analysis of patient samples in the liver hepatocellular carcinoma (LIHC) dataset in The Cancer Genome Atlas shows that high expression of one or more of the transcripts encoding the 10 components of the NUP107–160 subcomplex, which includes *AHCTF1*, is positively correlated with worse overall survival. These results provide a strong and feasible rationale for the development of novel cancer therapeutics that target ELYS function and suggest potential avenues for effective combinatorial treatments.

## Editor's evaluation

We believe that the study demonstrates the importance of nuclear pore complex components for Kras/p53 driven liver tumors. The findings made here in zebrafish may stimulate additional preclinical and mechanistic studies to test the role of nuclear pore components in cancer.

## Introduction

Synthetic lethality describes the death of cells in response to individual mutations in two separate genes, neither of which is lethal alone. The phenomenon has emerged as a promising framework for cancer drug development (*Gao and Lai, 2018*). Inherent to the approach is the capacity to induce the death of a vulnerable cell population, such as oncogene-expressing cancer cells, and leave healthy cells unaffected. In the clinic, the use of poly(adenosine diphosphate [ADP]-ribose) polymerase (PARP) inhibitors to successfully treat tumours carrying mutations in the breast cancer susceptibility genes, *BRCA1/BRCA2* (*Lord and Ashworth, 2017*), has validated the approach and driven the search for other clinically relevant gene pairings, including those that confer synthetic lethality in cancer cells expressing oncogenic mutations in *KRAS* (*Luo et al., 2009*; *Wang et al., 2017*). In this paradigm, the interacting gene is neither mutated nor oncogenic in its own right. Rather, its function is essential to maintain the tumourigenic state, inspiring the concept of non-oncogene addiction (*Solimini et al., 2007*).

In this study, we tested whether *AHCTF1* exhibits the properties of a synthetic lethal interacting gene with mutant *KRAS*. We became interested in *AHCTF1* when we and others showed that homozygous inheritance of an ENU-induced nonsense mutation in the zebrafish *ahctf1* gene (*ahctf1^{ti262}*) (*de Jong-Curtain et al., 2009*; *Davuluri et al., 2008*) disrupted nuclear pore formation and caused catastrophic levels of cell death in the intestinal epithelium and other highly proliferative cell compartments during zebrafish development (*de Jong-Curtain et al., 2009*; *Davuluri et al., 2008*). Meanwhile, cells in relatively quiescent tissues survived and remained healthy.

*AHCTF1* encodes ELYS, a 252-kDa multidomain nucleoporin (NUP) that was first discovered in mice where it was shown to be required for the proliferation and survival of inner mass cells during embryonic development (*Okita et al., 2004*). ELYS is one of 10 components of the large NUP107–160 subunit of nuclear pore complexes (NPCs). These huge (110 MDa) multi-subunit complexes comprise approximately 34 different NUPs in octameric array (*Beck and Hurt, 2017*; *Petrovic et al., 2022*), forming cylindrical channels in the nuclear envelope that regulate nucleocytoplasmic transport and intracellular localisation of large (>40 kDa) molecules. ELYS is also indispensable for NPC reassembly after mitosis (*Rasala et al., 2006*; *Gillespie et al., 2007*; *Franz et al., 2007*) and carries out a broad range of activities during the cell cycle, including chromatin decompaction, mitotic spindle assembly, and chromosome segregation (*Gillespie et al., 2007*; *Kuhn et al., 2019*; *Güttinger et al., 2009*; *Chatel and Fahrenkrog, 2011*; *Mishra et al., 2010*; *Yokoyama et al., 2014*; *Kobayashi et al., 2019*; *Rasala et al., 2008*).

Having shown previously that cells that are rapidly growing and dividing during zebrafish development are highly vulnerable to ELYS depletion, we hypothesised that cancer cells fuelled by powerful oncogenes would be vulnerable too. To test this, we took advantage of a genetically tractable zebrafish model of hepatocellular carcinoma (HCC) in which a doxycycline-inducible, hepatocyte-specific *EGFP-kras^{G12V}* transgene drives hepatocyte hyperplasia, liver enlargement, and morphological changes characteristic of human HCC (*Chew et al., 2014*). We chose this model of HCC because the RAS/RAF/MAPK signalling pathway is almost always hyperactivated in human HCC (*Calvisi et al., 2006*).

We found that reducing the expression of *ahctf1* mRNA by 50% disrupted multiple functions in *kras^{G12V}*-expressing hepatocytes and markedly impaired their growth and survival. Further studies showed that accumulation of DNA damage and robust Tp53 activation contributed to this response. These findings suggest that *ahctf1* and mutant *kras* participate in a synthetic lethal interaction that is selective for Kras oncogene-expressing cells, providing a rationale to investigate whether ELYS function could be targeted effectively and selectively by a new class of anti-cancer drugs.

## Results

### Molecular characterisation of mutant *kras*-driven zebrafish models of HCC

To generate both larval and adult models of HCC (*Chew et al., 2014*), we varied the timing of doxycycline (dox) treatment. In our larval model, on a wildtype (WT) *ahctf1* background, we induced the expression of a single *EGFP-kras$^{G12V}$* transgene, denoted *TO(kras$^{G12V}$)$^{T/+}$* in developing livers by treating with dox between 2 and 7 days post-fertilisation (dpf) (*Figure 1a, b*). This led to the accumulation of a constitutively active, EGFP-tagged, potently oncogenic form of Kras (Kras$^{G12V}$) specifically in hepatocytes, causing hepatocyte hyperplasia and a substantial increase (4-fold) in liver volume (*Figure 1c, d*). To establish the clinical relevance of this phenotype to human HCC, we used RNA sequencing to analyse the gene expression patterns of livers expressing the *kras$^{G12V}$* transgene compared to livers expressing no transgene. We detected more than 6000 significantly upregulated genes in dox-treated *TO(kras$^{G12V}$)$^{T/+}$* livers compared WT livers, and a further 6000+ genes were significantly downregulated (*Figure 1—figure supplement 1a, b*). Gene set enrichment analysis identified a positive correlation between the differential gene expression data from the dox-treated *TO(kras$^{G12V}$)$^{T/+}$* versus WT livers and the differential gene expression data obtained from the HCC (LIHC) and healthy liver subsets available in The Cancer Genome Atlas (TCGA) (*Figure 1—figure supplement 1c*). We also found a positive correlation between the diffentially expressed genes (DEGs) from the dox-treated *TO(kras$^{G12V}$)$^{T/+}$* versus WT livers and a small HCC expression signature based on four patient samples carrying KRAS G12 or KRAS G13 mutations (*Figure 1—figure supplement 1d*). Of the upregulated genes, many were significantly enriched in KEGG pathways associated with highly proliferative cancers, including DNA replication, cell cycle regulation, and DNA damage repair (*Figure 1—figure supplement 1e*). These observations build on previous reports that dox-treated *TO(kras$^{G12V}$)$^{T/+}$* zebrafish provide an authentic model of human HCC (*Zheng et al., 2014b*; *Huo et al., 2019*).

### *ahctf1* heterozygosity reduces liver overgrowth in a zebrafish model of *kras$^{G12V}$*-driven HCC

We investigated the requirement for Elys in this in vivo cancer setting by introducing a mutant *ahctf1* allele (*flo$^{ti262}$*) (*Chen et al., 1996*) into the genome of the *TO(kras$^{G12V}$)$^{T/+}$* model. This produced a 57% reduction in *ahctf1* mRNA expression in *ahctf1$^{+/-}$* larvae at 7 dpf, compared to larvae expressing WT *ahctf1* (*Figure 1b*). This is consistent with the nonsense mutation in *ahctf1$^{ti262}$* triggering nonsense mediated decay of mRNA transcribed from the affected allele, rather than it being translated and expressed as a truncated Elys protein. As a control for our experiments, we used another transgenic line, denoted *2-CLiP* (2-Colour Liver Pancreas), in which hepatocytes express dsRed fluorescence constitutively but no oncogenic transgene (*Korzh et al., 2008*). On this background, heterozygous (HET) *ahctf1$^{ti262}$* zebrafish develop normally (*Figure 1—figure supplement 2a*), reach sexual maturity, and exhibit a normal lifespan, as do HET *ahctf1* mice (*Okita et al., 2004*). Similarly, hepatocytes on the *TO(kras$^{G12V}$)$^{T/+}$* background receiving no dox treatment developed normally (*Figure 1—figure supplement 2b*). Mean liver volume in control *2-CLiP* larvae at 7 dpf was $1.95 \times 10^6 \pm 4.99 \times 10^4$ μm$^3$ and was unaffected by *ahctf1* genotype (*Figure 1—figure supplement 3a, b*).

Dox-induced expression of oncogenic Kras$^{G12V}$ in the *TO(kras$^{G12V}$)$^{T/+}$* model, produced a striking (4-fold) increase in liver volume ($7.97 \times 10^6 \pm 1.21 \times 10^5$ μm$^3$) over the 5 days of dox treatment (*Figure 1c, d*). Remarkably, liver volume was pared back to $5.92 \times 10^6 \pm 8.83 \times 10^4$ μm$^3$ in *ahctf1* HETS, equating to a 35% reduction in excess liver volume. *ahctf1* heterozygosity also reduced liver enlargement in *kras$^{G12V}$*-expressing adult zebrafish. To induce HCC in these experiments, sexually mature zebrafish aged 3 months post-fertilisation were treated with dox (final concentration 20 mg/L for 7 days with fresh water and dox administered daily. After 7 days, the impact of forced *kras$^{G12V}$* expression was assessed in both male and female adults by measuring the mass of the liver and expressing this as a percentage of the total mass of the animal prior to liver dissection (*Figure 1e*). Compared to vehicle-treated animals, we found that dox induction of *kras$^{G12V}$* expression produced a robust (9-fold) increase in liver mass expressed as a percentage of total body mass of male zebrafish expressing WT *ahctf1*. This ratio was reduced by 28% in *ahctf1* HETS and we obtained similar results with females (*Figure 1e*). Histological sections of livers from vehicle-treated *ahctf1$^{+/+}$; TO(kras$^{G12V}$)$^{T/+}$* males stained with haematoxylin and eosin (H&E) revealed an orderly arrangement of polygonal hepatocytes

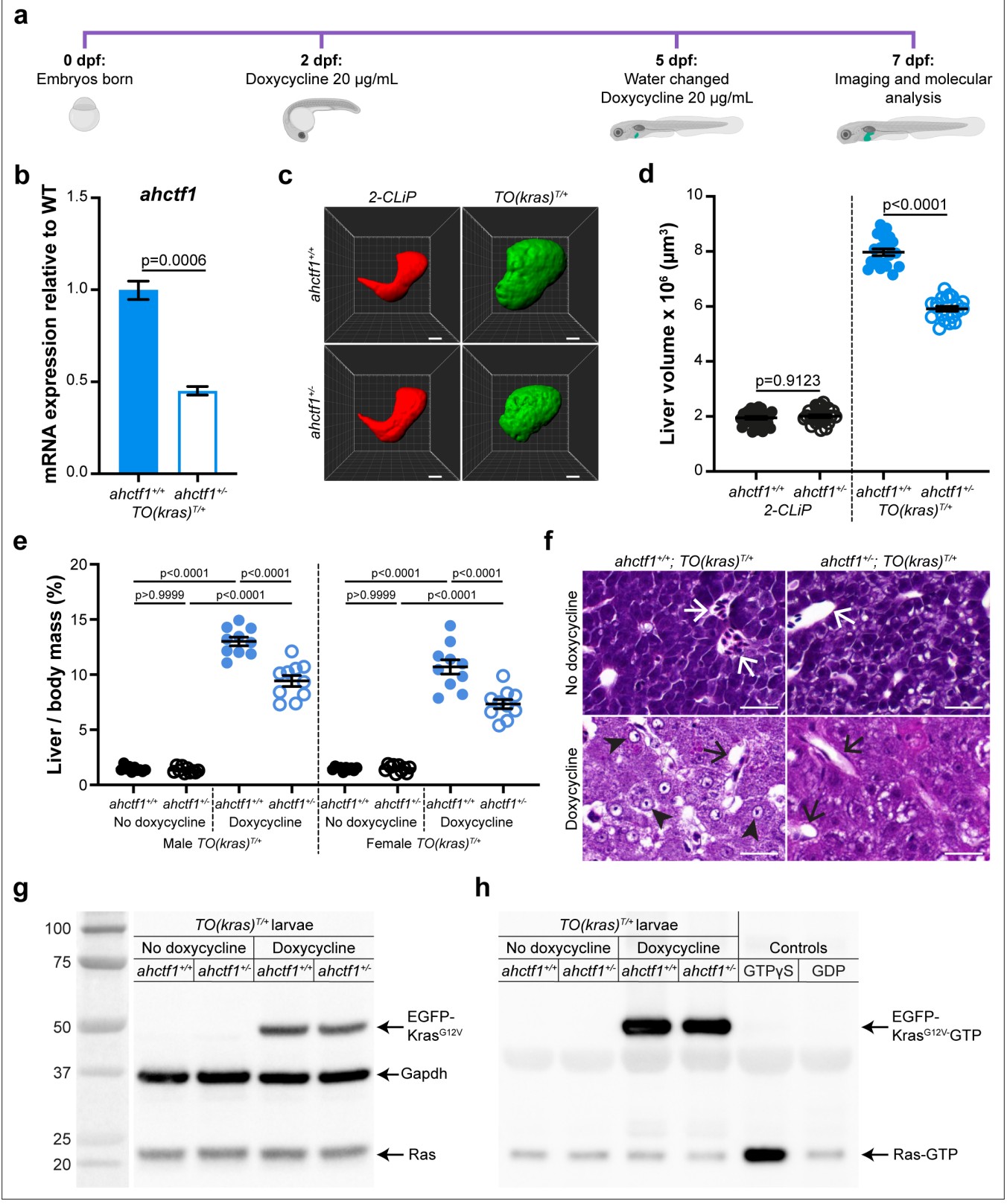

**Figure 1.** *ahctf1* heterozygosity restricts liver volume in a zebrafish model of *kras*$^{G12V}$-driven hepatocellular carcinoma (HCC). (**a**) Protocol used to induce *TO(kras*$^{G12V}$*)*$^{T/+}$ expression in the livers of developing zebrafish larvae. (**b**) RT-quantitative PCR (RT-qPCR) analysis of *ahctf1* mRNA levels in pooled micro-dissected larval livers (*n* = 3 biological replicates). (**c**) Representative three-dimensional reconstructions of *2-CLiP* and dox-treated *TO(kras*$^{G12V}$*)*$^{T/+}$ larval livers of the indicated *ahctf1* genotype. Scale bar 25 µm. (**d**) Impact of *ahctf1* heterozygosity on liver volume in *2-CLiP* and *TO(kras*$^{G12V}$*)*$^{T/+}$ larvae (*n* ≥ 20).

*Figure 1 continued on next page*

*Figure 1 continued*

(**e**) Impact of *ahctf1* heterozygosity on liver-to-body mass ratio of adult *TO(kras^G12V^)^T/+^* zebrafish in the presence or absence of dox treatment (*n* = 10). (**f**) Histological sections of adult male *TO(kras^G12V^)^T/+^* zebrafish livers of the indicated *ahctf1* genotype and dox treatment, stained with haematoxylin and eosin. In vehicle-treated adults, hepatocytes are densely packed and well differentiated. White arrows point to sections through blood vessels containing red blood cells. Meanwhile, the hepatocytes in dox-treated animals are poorly differentiated and exhibit multiple cytological abnormalities, including pyknotic nuclei (arrowheads) and vacuolation (black arrows). Scale bar 25 µm. (**g**) Western blot of Ras and Gapdh protein signals in total input lysates (50 µg) of *TO(kras^G12V^)^T/+^* larvae of the indicated *ahctf1* genotype and dox treatment. (**h**) Western blot of active Ras-GTP protein signals in lysates following active Ras pull-down.

The online version of this article includes the following source data and figure supplement(s) for figure 1:

Source data 1. *Figure 1b*: Heterozygous *ahctf1* mRNA expression in the livers of doxycycline (dox)-treated *TO(kras^G12V^)^T/+^* zebrafish larvae, relative to wildtype (WT).

Source data 2. *Figure 1g*: Uncropped unlabelled western blot.

Source data 3. *Figure 1g*: Uncropped western blot (labelled).

Source data 4. *Figure 1h*: Uncropped unlabelled western blot.

Source data 5. *Figure 1h*: Uncropped western blot (labelled).

Figure supplement 1. Analysis of RNAseq data from livers micro-dissected from *TO(kras^G12V^)^+/+^* (wildtype, WT) and dox-treated *TO(kras^G12V^)^T/+^* larvae reveals patterns of differential gene expression that recapitulate human hepatocellular carcinoma (HCC).

Figure supplement 2. *ahctf1* heterozygosity and/or homozygous *tp53* mutation do not impact on body length.

Figure supplement 3. *ahctf1* heterozygosity and/or homozygous *tp53* mutation do not impact on liver volume during normal liver development.

interspersed with blood vessels containing red blood cells. In the absence of dox, this appearance was unaffected by *ahctf1* genotype (*Figure 1f*, top row). In contrast, dox-treated *ahctf1^+/+^; TO(kras^G12V^)^T/+^*-expressing male livers exhibited severely disrupted architecture, and a general loss of hepatocyte organisation (*Figure 1f*, bottom left). Hepatocytes were generally poorly differentiated and some exhibited features such as pyknotic nuclei, condensed nucleoli, and vacuolation (*Figure 1f*, bottom left). Sections of *kras^G12V^*-expressing livers from *ahctf1* HETS showed hepatocytes with improved nuclear integrity and less vacuolation (*Figure 1f*, bottom right). These data show that in both larval and adult *TO(kras^G12V^)^T/+^*-expressing zebrafish, a modest (50%) decrease in *ahctf1* mRNA expression exerts a robust and selective reduction in hepatocyte hyperplasia and liver overgrowth.

## Level of activated (GTP-bound) Ras in dox-treated *TO(kras^G12V^)^T/+^* larvae is unaffected by *ahctf1* heterozygosity

To determine how heterozygous *ahctf1* mRNA expression restricts mutant *kras*-driven liver overgrowth, we investigated whether the Elys protein could interfere with the activation of Kras^G12V^ directly. To do this, we first quantitated the abundance of the EGFP-Kras^G12V^ protein using western blot analysis. Dox treatment markedly increased the abundance of the EGFP-Kras^G12V^ protein in lysates of *TO(kras^G12V^)^T/+^* zebrafish larvae compared to vehicle-treated larvae (*Figure 1g*), and this was unaffected by *ahctf1* genotype. We then used an active (GTP-bound) Ras pull-down assay to isolate the active GTP-bound fraction of Ras proteins in lysates of *TO(kras^G12V^)^T/+^* larvae, followed by western blot analysis (*Baker and Rubio, 2021*). In vehicle-treated zebrafish larvae, signals corresponding to GTP-bound endogenous Ras proteins were weak, whereas lysates from dox-treated larvae gave robust signals corresponding to activated EGFP-Kras^G12V^ proteins. The intensity of these signals was not affected by *ahctf1* genotype (*Figure 1h*). Thus, the impact of heterozygous *ahctf1* mutation on the growth, proliferation, and survival of *TO(kras^G12V^)^T/+^* expressing hepatocytes did not occur by directly or indirectly interfering with the production of GTP-bound Ras proteins.

## *ahctf1* heterozygosity disrupts the abundance of NPCs in dox-treated *TO(kras^G12V^)^T/+^* hepatocytes

Having demonstrated that *ahctf1* heterozygosity restricts *kras^G12V^*-driven liver enlargement, we sought to understand the biological mechanisms underlying this. As previously mentioned, Elys is a multi-functional protein with several roles in the cell cycle (*Figure 2—figure supplement 1*). We started by examining whether its canonical role in post-mitotic nuclear pore assembly (*Figure 2—figure supplement 1a, b*) was disrupted. To measure the abundance and distribution of NPCs, we stained

thick sections of larval livers (200 μm), with an antibody (mAb414) that recognises FG-repeat NUPs (NUP358, 214, 153, and 62) in mature NPCs. Using Airyscan confocal laser-scanning microscopy, we showed that hepatocytes not carrying the *TO(kras^G12V)* transgene (denoted *TO(kras)^+/+*), exhibit fluorescent puncta corresponding to NPCs at the nuclear rim with negligible staining in the cytoplasm, a pattern that was unaffected by *ahctf1* genotype (*Figure 2a*; left two columns). In dox-treated larvae harbouring the *TO(kras^G12V)^T/+* transgene, fluorescence intensity was markedly increased (*Figure 2a*, third column) and there was an increase in the ratio of nuclear:cytoplasmic staining (*Figure 2b*). By comparison, fluorescence intensity at the nuclear rim was diminished in *ahctf1* HETS, concomitant with the appearance of fluorescent puncta in the cytoplasm (*Figure 2a*, fourth column; arrows) and a significant reduction in the ratio of nuclear:cytoplasmic fluorescence intensity (*Figure 2b*).

To determine the abundance of NPCs, we analysed the pattern and density of fluorescent puncta observed at the nuclear surface of non-*kras^G12V* expressing hepatocytes (*Figure 2c*; left two columns). The induced expression of the *kras^G12V* transgene in the presence of WT *ahctf1* resulted in 59% more fluorescent puncta/NPCs at the nuclear surface of hyperplastic hepatocytes (*Figure 2c, d*; third column). These signals were reduced by 21% in *ahctf1* HETS (*Figure 2c, d*; fourth column). Induced *kras^G12V* expression also produced a 28% increase in nuclear volume compared to non-*kras^G12V*-expressing cells (*Figure 2e*), and this increase in size was reduced to 13% in *ahctf1* HETS. We infer from these data that hyperplastic hepatocytes expressing the *kras^G12V* oncogene require highly efficient rates of nucleocytoplasmic transport to support their rapid proliferation, which they fulfil by increasing NPC density and the size of their nuclei. This adaptation required a full complement of *ahctf1* expression, suggesting non-oncogene addiction to *ahctf1*. We found that in hyperplastic hepatocytes that were heterozygous for *ahctf1* this adaptation was partially restricted, creating a condition likely to amplify oncogenic stress.

## *ahctf1* heterozygosity impairs mitotic spindle assembly and chromosome segregation in dox-treated *TO(kras^G12V)^T/+* hepatocytes

Next, we examined the impact of reduced Elys expression on spindle formation and chromosome segregation during mitosis (*Figure 2—figure supplement 1g and h*). We assessed these features in cryosections of liver using α-tubulin and 4',6-diamidino-2-phenylindole (DAPI) to stain microtubules and chromatin, respectively. Metaphase cells in *ahctf1^+/+;TO(kras^G12V)^T/+* livers exhibited normal bipolar spindle formation followed by complete chromosome segregation during anaphase (*Figure 3a*). In contrast, metaphase cells in *ahctf1^+/−;TO(kras^G12V)^T/+* hepatocytes displayed abnormal multipolar spindles and misaligned chromosomes (*Figure 3b*). Proper chromosome segregation was disrupted with multiple anaphase bridges formed. While the number of cells observed at different mitotic stages was similar in *ahctf1^+/+* and *ahctf1^+/−* (*Figure 3c*), mitotic abnormalities were observed in 50% of *ahctf1^+/−;TO(kras^G12V)^T/+* hepatocytes during metaphase and anaphase but not at all in *ahctf1^+/+;TO(kras^G12V)^T/+* hepatocytes (*Figure 3d*). These data are consistent with hyperplastic hepatocytes requiring a full complement of *ahctf1* expression to maintain rapid rounds and integrity of mitosis, and provide another facet of non-oncogene addiction to *ahctf1* that is likely to contribute to oncogenic stress.

## *ahctf1* heterozygosity causes DNA damage and accumulation of Tp53 protein in dox-treated *TO(kras^G12V)^T/+* hepatocytes

Other features of oncogene-induced stress in response to robust and persistent overexpression of RAS oncoproteins include stalled DNA replication, DNA damage, and genome instability. To determine whether expression of *kras^G12V* causes DNA damage in our model, we stained cryosections of larval livers with DAPI and γ-H2AX, which is a sensitive marker for stalled DNA replication forks and DNA double-strand breaks (*Rogakou et al., 1998*). We found that 1% of *ahctf1^+/+;TO(kras^G12V)^T/+* hepatocyte nuclei were positive for γ-H2AX (*Figure 4a, c*), compared to 6% in the nuclei of *ahctf1* HETS (*Figure 4b, c*).

In the presence of WT Tp53, DNA damage is limited by activation of Tp53 transcription-dependent pathways that can perform DNA damage repair, and, if necessary, induce cell cycle arrest, senescence, and/or apoptosis. To determine whether the increase in DNA damage that occurred in our model in response to *ahctf1* heterozygosity stimulated Tp53 accumulation, we measured the levels of Tp53 protein in pooled lysates of micro-dissected *kras^G12V*-expressing livers, *kras^G12V*-nonexpressing

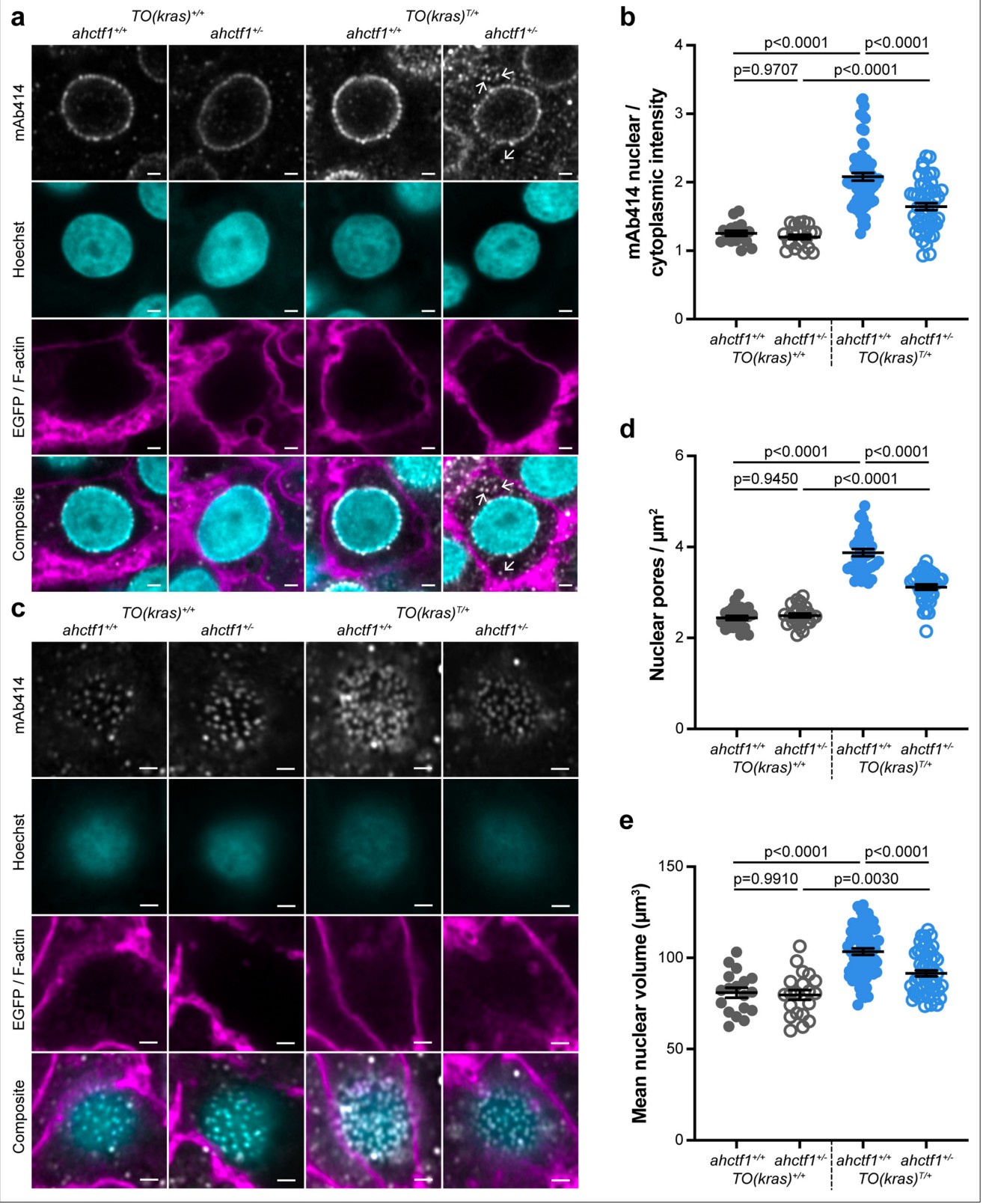

**Figure 2.** *ahctf1* heterozygosity disrupts the density of nuclear pore complexes and reduces nuclear volume in dox-treated *TO(kras^{G12V})^{T/+}* hepatocytes. (**a**) Representative Airyscan imaging of liver sections stained with mAb414 (white) marking FG-NUPs, Hoechst 33342 (cyan) marking DNA, and rhodamine phalloidin (magenta) marking the F-actin cytoskeleton in non-*TO(kras^{G12V})*-expressing cells and EGFP-Kras^{G12V} (magenta) marking the cell membrane in dox-treated *TO(kras^{G12V})*-expressing cells of the indicated *ahctf1* and *TO(kras^{G12V})* genotypes. Arrows in right-hand panel point to mAb414/FG-

*Figure 2 continued on next page*

*Figure 2 continued*

nucleoporin staining in the cytoplasm. Scale bar 2 µm. (**b**) Quantification of mean nuclear/cytoplasmic fluorescence intensity of mAb414 staining after 3D segmentation and morphological filtering of nuclear and cytoplasmic areas ($n > 18$). (**c**) Representative Airyscan images of mAb414 staining at the nuclear surface of sections of the indicated *ahctf1* and *TO(kras^G12V)* genotype. Scale bar 1 µm. (**d**) Quantification of nuclear pore density ($n \geq 25$). (**e**) Quantification of nuclear volume ($n \geq 25$). Data are expressed as mean ± standard error of the mean (SEM). Significance was assessed using a one-way analysis of variation (ANOVA) with Tukey's multiple comparisons test.

The online version of this article includes the following source data and figure supplement(s) for figure 2:

**Source data 1.** *Figure 2b*: Ratio of nuclear to cytoplasmic intensity of mAb414 immunostaining in livers of dox-treated *TO(kras^G12V)^T/+* zebrafish larvae; role of *ahctf1* genotype.

**Figure supplement 1.** ELYS (encoded by *AHCTF1*) is a protein component of nuclear pore complexes (NPCs) with multiple functions during the cell cycle.

---

livers and the larval remains after the dissection (*Figure 4d, e*). No Tp53 signal was obtained from non-*kras^G12V*-expressing livers, or the body remains after liver removal. However, we detected a weak Tp53 signal in extracts of dox-treated *ahctf1^+/+;TO(kras^G12V)^T/+* livers, consistent with induced *kras^G12V* expression causing mild cellular stress. This low level of stress was amplified significantly in heterozygous *ahctf1* livers, where we obtained a strong (>3.5-fold) increase in the intensity of the Tp53 signal (*Figure 4e*), which correlated well with the higher density of γ-H2AX staining.

To test whether Tp53 accumulation was responsible for limiting liver volume in *TO(kras^G12V)^T/+* larvae, we introduced the zebrafish *tp53^M214K* allele encoding a transactivation dead Tp53 variant (*Berghmans et al., 2005*). Abrogating Tp53 function by homozygous expression of this allele (denoted *tp53^m/m*) in *ahctf1^+/+;TO(kras^G12V)^T/+* larvae increased liver volume by 50%, to $12.5 \times 10^6 \pm 1.10 \times 10^5$ µm³, compared to livers on a WT (*tp53^+/+*) background (*Figure 4f, g*), demonstrating that Tp53 function normally places great restraint on the growth of *TO(kras^G12V)^T/+* livers in this model. *ahctf1* heterozygosity combined with loss of Tp53 function also produced an increase in liver volume by 50% compared to *ahctf1^+/−;TO(kras^G12V)^T/+* livers on a WT Tp53 background. Comparing the volume of *ahctf1* HET livers and WT *ahctf1* livers on a Tp53 mutant background, showed that heterozygous *ahctf1* still achieved a reduction in liver volume, albeit significantly less than in the presence of WT Tp53.

## *ahctf1* heterozygosity amplifies cell death of dox-treated *TO(kras^G12V)^T/+* hepatocytes in the presence and absence of WT Tp53

The tumour suppressive properties of Tp53 lie in its capacity to activate the transcription of genes that participate in processes that restrict tumour growth, including cell cycle arrest, senescence, apoptosis, DNA repair, and metabolic adaptation. To test whether disruption of these processes contributed to the reduction in liver volume we observe between WT and heterozygous *ahctf1* larvae, we first looked at apoptosis. To do this, we introduced an apoptosis reporter transgene, Tg(*actb2:SEC-Hsa. ANXA5-mKate2,cryaa:mCherry*)^uq24rp (hereafter denoted *Annexin 5-mKate*) into the HCC model. This transgene constitutively expresses a fusion protein comprising Annexin 5 and the far-red fluorophore mKate (*Hall et al., 2019*). The protein generates discrete fluorescent puncta in cells undergoing apoptosis by binding to phosphatidylserine molecules that become accessible to Annexin 5 binding when the plasma membrane breaks down during apoptosis. The abundance of Annexin 5 fluorescent puncta was 1.5 puncta per $10^{-5}$ µm³ in the livers of WT Tp53, *ahctf1^+/+;TO(kras^G12V)^T/+* larvae and this value increased by 3.5-fold (5.2 puncta per $10^{-5}$ µm³) on a heterozygous *ahctf1* background (*Figure 5a*, left two columns, *Figure 5b*). We saw a comparable increase in the number of apoptotic cells in heterozygous *ahctf1* livers when we used the cleaved (active) form of caspase-3 as an alternative marker of apoptosis (*Figure 5—figure supplement 1*).

To determine whether Tp53 accumulation played a role in this cell death, we measured the density of Annexin 5 puncta in the absence of functional Tp53. Compared to those expressing WT Tp53 (1.5 puncta per $10^{-5}$ µm³), Annexin 5 puncta were relatively sparse (0.13 puncta per $10^{-5}$ µm³) in Tp53-deficient livers on a *ahctf1^+/+;TO(kras^G12V)^T/+* background. This robust (11.3-fold) reduction in the density of Annexin 5 foci indicates that the death of hepatocytes in *ahctf1* WT, *TO(kras^G12V)^T/+* livers is normally kept in check by Tp53 (*Figure 5a, b*). Turning to *ahctf1* HETS, the density of Annexin 5 puncta on a *tp53^m/m* background was 2.3 puncta per $10^{-5}$ µm³, less than half the number of puncta (5.2 per $10^{-5}$ µm³) on a WT Tp53 background (*Figure 5a, b*). The number of hepatocytes undergoing apoptosis on a Tp53-deficient background remained higher in heterozygous *ahctf1* larvae compared

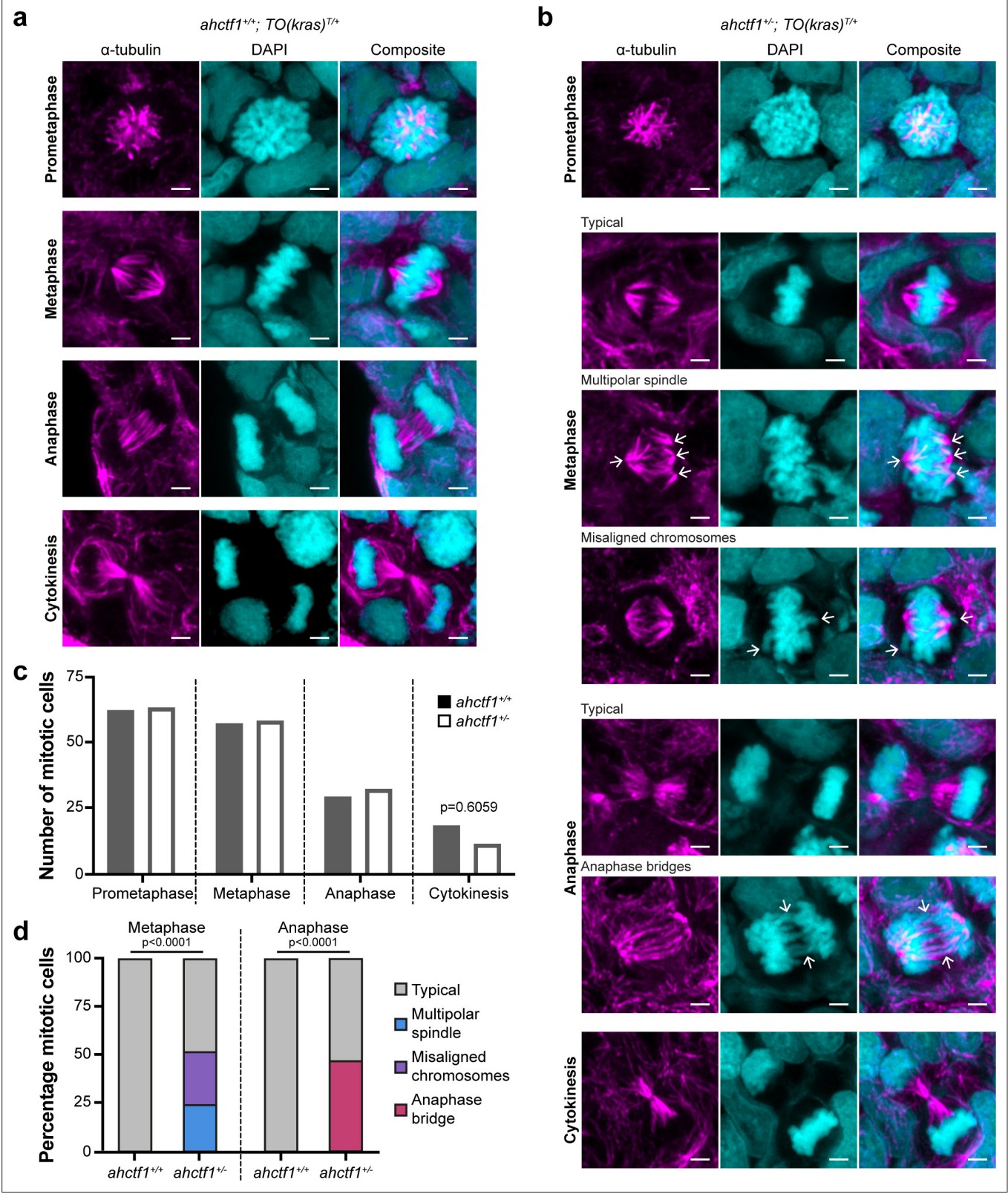

**Figure 3.** *ahctf1* heterozygosity impairs mitotic spindle assembly and chromosome segregation in dox-treated *TO(kras^{G12V})^{T/+}* hepatocytes. (**a**) Representative Airyscan imaging of liver cryosections stained with α-tubulin antibody (magenta) marking spindle microtubules and DAPI (cyan) marking DNA in mitotic cells of *TO(kras^{G12V})^{T/+}* larvae on a wildtype *ahctf1^{+/+}* background. (**b**) Mitotic cells in liver cryosections of *TO(kras^{G12V})^{T/+}* larvae that are heterozygous for *ahctf1^{+/−}* exhibit multiple defects, including multipolar spindles, misaligned chromosomes, and anaphase bridges (arrows). Scale

*Figure 3 continued on next page*

*Figure 3 continued*

bar 2 μm. (**c**) Distribution of cells observed at different mitotic stages (*n* = 92 livers, 326 mitotic cells). (**d**) Quantification of the percentage of mitotic hepatocytes exhibiting an aberrant phenotype (*n* = 14–57). Significance was assessed using a Chi-square test.

The online version of this article includes the following source data for figure 3:

**Source data 1.** *Figure 3c*: Number of cells at different stages of mitosis in livers of *TO(krasG12V)T/+* zebrafish larvae; role of *ahctf1* genotype.

to *ahctf1* WT larvae, albeit it was much reduced compared to that observed in the presence of WT Tp53.

To determine whether these responses were tied to the expression of pro-apoptotic Tp53 target genes in *krasG12V*-driven hepatocytes, we used RT-quantitative PCR (RT-qPCR) analysis of pooled micro-dissected livers (*Figure 5c*). The abundance of *tp53* mRNA was unaffected by *ahctf1* heterozygosity (*Figure 5—figure supplement 2a*), which was to be expected because the accumulation of Tp53 protein is regulated post-translationally. However, we could infer that Tp53 transcription was activated by *ahctf1* heterozygosity by the expression of the canonical Tp53 target genes, *mdm2* and *Δ113tp53*, which were upregulated by 4.7-fold (*Figure 6c*) and >6-fold, respectively (*Figure 5—figure supplement 2b*), compared to when *ahctf1* was WT.

We then examined the expression levels of Bcl2 family genes that regulate mitochondrial apoptosis including mRNAs encoding the six BH3-only apoptosis effector proteins, Pmaip1 (aka Noxa) and Bbc3 (aka Puma), Bim, Bid, Bik, and Bad, and the pro-apoptotic executioner protein, Bax (*Figure 5c*). All were significantly increased in the livers of *ahctf1* HETS compared to WT *ahctf1* livers. The most marked upregulation was observed with the classical Tp53 transcriptional targets, *pmaip1*, *bbc3*, and *bax* (*Figure 5c*). In contrast, transcripts encoding the pro-survival proteins, Bcl2 and Bclxl, were significantly downregulated by *ahctf1* heterozygosity (*Figure 5c*). In the absence of Tp53 function, *ahctf1* heterozygosity produced more modest increases in the expression of the pro-apoptotic transcripts and smaller decreases in the expression of the pro-survival *bcl2* and *bclxl* genes (*Figure 5c*). We surmise from these data that *ahctf1* heterozygosity contributes to a reduction in liver overgrowth in *TO(krasG12V)T/+* larvae by activating a largely Tp53-dependent transcriptional programme that promotes apoptosis.

## *ahctf1* heterozygosity restricts DNA replication in dox-treated *TO(krasG12V)T/+* hepatocytes

To determine whether DNA replication and cell cycle progression were affected by *ahctf1* heterozygosity (*Figure 2—figure supplement 1c–e*), we used an EdU incorporation assay to identify the number of DNA replicating cells in S-phase (*Figure 6a*). We found that 32% of hyperplastic hepatocytes were positive for EdU in *ahctf1+/+;TO(krasG12V)T/+* larvae on a WT Tp53 background (*Figure 6b*) and this abundance was reduced to 22% of hepatocytes in *ahctf1+/−;TO(krasG12V)T/+* larvae (*Figure 6b*). To determine whether Tp53 played a role in the reduced frequency of cells in S-phase, we repeated the experiment in Tp53-deficient larvae. This resulted in a striking increase in the number of cells in S-phase whereby 63% of hepatocytes were EdU positive in WT *ahctf1* larvae, and 53% of hepatocytes were EdU positive in *ahctf1* HETS (*Figure 6b*). These data show that *ahctf1* heterozygosity reduces the number of cells in S-phase by a mechanism that largely depends on the availability of WT Tp53. The changes in EdU incorporation between hepatocytes of different *ahctf1* and *tp53* genotypes were accompanied by corresponding changes in liver volume (*Figure 6c*).

Next, we determined whether the expression of Tp53-dependent negative cell cycle regulators was involved in the reduced number of hepatocytes in S-phase we observed in *ahctf1* HETS. We saw a 3-fold increase in the expression of *cdkn1a* and a 2.5-fold increase in the expression of *cdkn2a/b* in *ahctf1* HET larvae compared to larvae that were WT for *ahctf1* (*Figure 6d*). The upregulation of *cdkna1* expression in *ahctf1* HETS was partially reduced by Tp53 deficiency, suggesting that the regulation of *cdkna2* expression was largely, but not completely, Tp53 dependent. In contrast, the upregulation of *cdkna2a/b* was entirely lost in *ahctf1* HETS, showing that in this setting, *cdkna2a/b* upregulation was dependent on the availability of functional Tp53 (*Figure 6d*).

We also found that mRNAs encoding the Tp53 family members, Tp63 and Tp73 were upregulated by 3.0- and 2.3-fold, respectively, in *ahctf1* HET livers, compared to WT *ahctf1* livers, in the presence of WT Tp53 (*Figure 6e*). Interestingly, in the absence of WT Tp53 (*Figure 6e*), *ahctf1* heterozygosity

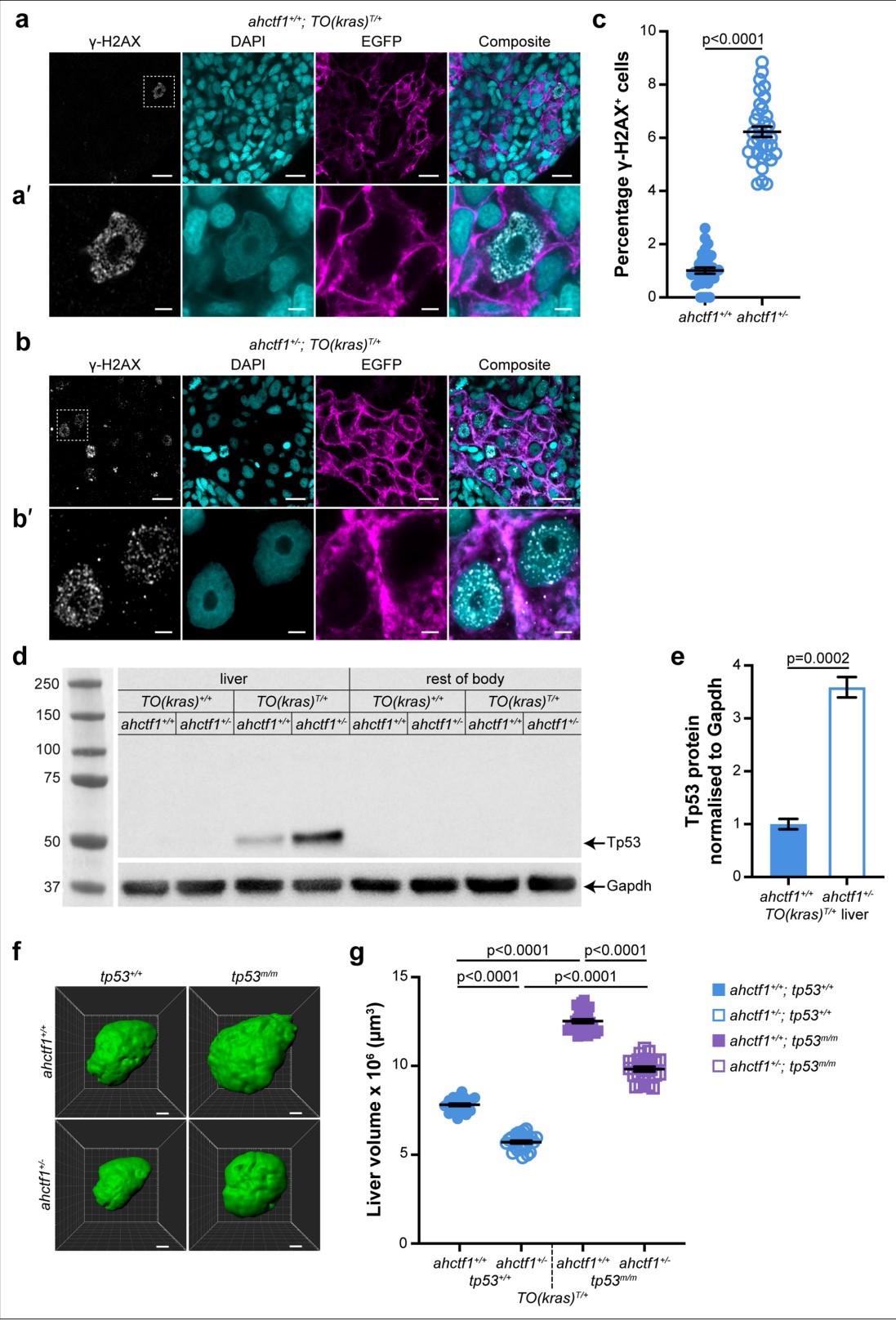

**Figure 4.** *ahctf1* heterozygosity causes DNA damage and accumulation of Tp53 protein in dox-treated *TO(kras^{G12V})^{T/+}* hepatocytes. (**a**) Representative Airyscan imaging of cryosections of liver from dox-treated *ahctf1^{+/+};TO(kras^{G12V})^{T/+}* larvae stained with γ-H2AX antibody (white) marking DNA double-strand breaks, DAPI (cyan) marking DNA and EGFP-Kras^{G12V} (magenta) marking the cell membrane. Scale bar 5 μm. (**a′**) Inset of γ-H2AX-positive nuclei in *ahctf1^{+/+};TO(kras^{G12V})^{T/+}* hepatocytes. Scale bar 2 μm. (**b**) Representative images of cryosections of liver from dox-treated *ahctf1^{+/-};TO(kras^{G12V})^{T/+}*

*Figure 4 continued on next page*

# eLife Research article

*Figure 4 continued*

larvae. Scale bar 5 µm. (**b'**) Inset of γ-H2AX-positive nuclei in cryosections of liver from *ahctf1⁺/⁻;TO(kras^G12V)^T/+* larvae. Scale bar 2 µm. (**c**) Quantification of the percentage of hepatocytes positive for γ-H2AX (*n* ≥ 31). (**d**) Representative western blot of Tp53 protein signals in lysates of *TO(kras^G12V)* larvae of the indicated *ahctf1* genotype. (**e**) Quantification of Tp53 protein levels normalised by reference to the Gapdh loading control (*n* = 3 independent experiments). (**f**) Representative three-dimensional reconstructions of dox-treated *TO(kras^G12V)^T/+* livers of the indicated *ahctf1* and *tp53* genotypes. Scale bar 25 µm. (**g**) Impact of *ahctf1* heterozygosity and homozygous *tp53* mutation on liver volume in *2-CLiP* and *TO(kras^G12V)^T/+* larvae (*n* ≥ 20). Data are expressed as mean ± standard error of the mean (SEM). Significance was calculated using a Student's *t*-test or one-way analysis of variation (ANOVA) with Tukey's multiple comparisons test.

The online version of this article includes the following source data for figure 4:

**Source data 1.** *Figure 4c*: Percentage of hepatocytes positive for γ-H2AX immunostaining in livers of dox-treated *TO(kras^G12V)^T/+* zebrafish larvae; role of *ahctf1* genotype.

**Source data 2.** *Figure 4d*: Uncropped unlabelled western blot.

**Source data 3.** *Figure 4d*: Uncropped western blot (labelled).

caused a further, albeit mild, increase in the expression of *tp63* and *tp73*. Since *tp63* and *tp73* are not direct Tp53 target genes, this raises the possibility that elevated *tp63* and *tp73* expression may contribute modest Tp53-independent roles to the inhibition of liver overgrowth, activation of apoptosis and impaired cell cycle progression we observed in response to *ahctf1* heterozygosity. This is plausible because Tp63 and Tp73 have several direct target genes in common with Tp53, including the pro-apoptotic genes, *bbc3*, *pmaip1*, *bim*, and *bax*, and the cell cycle arrest gene, *cdkna1*.

We also examined the transcriptional profile of pro-apoptotic and cell cycle arrest genes in our adult model of HCC. Looking specifically at the expression of direct Tp53 target genes in response to *ahctf1* heterozygosity, we found that *Δ113tp53*, *mdm2*, *pmaip1*, *bbc3*, *ccgn1*, and *cdkn1a*, were all robustly upregulated in dox-treated *TO(kras^G12V)^T/+* livers (**Figure 6—figure supplement 1a–d**), as was *cdkn2a/b*, which is not a direct Tp53 target (**Figure 6—figure supplement 1e**). These data suggest that Tp53 plays a central role in mediating the reduction in liver enlargement conferred by heterozygous *ahctf1* expression in adult dox-treated *TO(kras^G12V)^T/+* livers, in agreement with our findings with the larval model of HCC.

Collectively, our data demonstrate that the reduction in liver overgrowth caused by *ahctf1* heterozygosity in dox-treated *TO(kras^G12V)^T/+* zebrafish is concurrent with the induction of largely Tp53-dependent transcriptional programmes that promote cell cycle arrest and apoptosis. However, we cannot infer from our data whether these two cellular responses are activated simultaneously in individual hepatocytes, or whether they are regulated in a dynamic fashion.

## Combinatorial targeting of *ahctf1* and *ranbp2* expression blocks liver overgrowth in dox-treated *TO(kras^G12V)^T/+* larvae

To extend our understanding of the role of NUPs in hyperplastic growth, we took advantage of a zebrafish mutant we had previously identified in a transgene assisted ENU-mutagenesis screen (**Ober et al., 2006**), which harboured a nonsense mutation in *ranbp2* (**Figure 7—figure supplement 1a, b**). Like several other digestive organ mutants identified in this screen (**de Jong-Curtain et al., 2009**; **Boglev et al., 2013**; **Markmiller et al., 2014**), *ranbp2^s452^* mutants exhibit morphological defects in proliferative compartments, including the intestinal epithelium, craniofacial complex, and eye (**Figure 7—figure supplement 1c, d**). RANBP2 (also known as NUP358) is a major component of the cytoplasmic filaments of NPCs, where it plays a role in nucleocytoplasmic trafficking. Like ELYS, RANBP2 also plays non-canonical roles beyond the NPC, including in mitotic progression and the maintenance of genome integrity (**Joseph et al., 2004**; **Salina et al., 2003**; **Hashizume et al., 2013**).

Like *ahctf1* heterozygosity, we found that *ranbp2* heterozygosity alone, or in combination with *ahctf1* heterozygosity, had no impact on liver volume in control (*2-CLiP*) larvae (**Figure 7a, b**). In *TO(kras^G12V)^T/+* larvae that were WT for both *ranbp2* and *ahctf1*, dox treatment caused a >3.8-fold increase in liver volume. Upon introduction of a single mutation of either *ahctf1* or *ranbp2* into this setting, the excess liver volume was pared back by 38% and 13%, respectively (**Figure 7b**). Remarkably, when we measured liver volume in *TO(kras^G12V)^T/+* larvae that were trans heterozygous for *ranbp2*

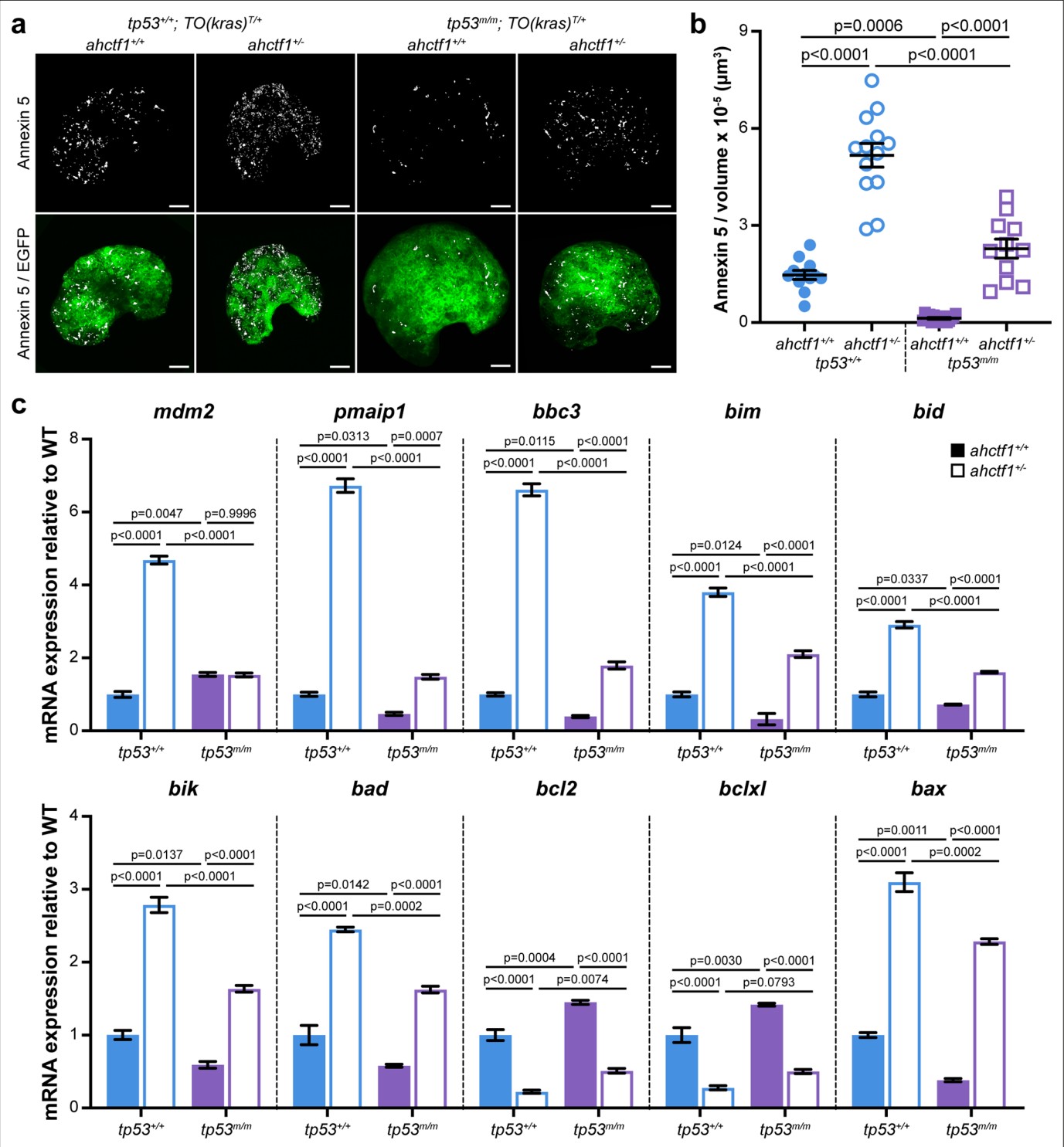

**Figure 5.** *ahctf1* heterozygosity amplifies cell death in dox-treated *TO(kras$^{G12V}$)$^{T/+}$* hepatocytes in the presence and absence of Tp53. (**a**) Representative maximum intensity projection images of Annexin 5-mKate fluorescence (white puncta), indicating cells undergoing apoptosis in *TO(kras$^{G12V}$)$^{T/+}$* livers of the indicated *ahctf1* and *tp53* genotypes. Scale bar 25 μm. (**b**) Quantification of the density of Annexin 5 fluorescent foci in *TO(kras$^{G12V}$)$^{T/+}$* livers of the indicated *ahctf1* and *tp53* genotypes (*n* ≥ 11). (**c**) RT-quantitative PCR (RT-qPCR) analysis of the specified mRNAs in *TO(kras$^{G12V}$)$^{T/+}$* micro-dissected livers of the indicated *ahctf1* and *tp53* genotypes (*n* = 3 biological replicates). Data are expressed as mean ± standard error of the mean (SEM). Significance was calculated using a one-way analysis of variation (ANOVA) with Tukey's multiple comparisons test.

The online version of this article includes the following source data and figure supplement(s) for figure 5:

*Figure 5 continued on next page*

*Figure 5 continued*

**Source data 1.** *Figure 5b*: Quantitation of Annexin 5-mKate fluorescence per volume (µm³) in livers of dox-treated *TO(kras^{G12V})^{T/+}* zebrafish larvae; role of *ahctf1* and *tp53* mutation.

**Figure supplement 1.** *ahctf1* heterozygosity increases the percentage of *TO(kras^{G12V})^{T/+}* hepatocytes expressing the cleaved, active form of caspase-3.

**Figure supplement 2.** *ahctf1* heterozygosity reduces *ahctf1* mRNA expression and increases expression of *Δ113tp53*.

and *ahctf1*, we found a striking, synergistic decrease in liver volume, which by 7 dpf, was not significantly different from that of *2-CLiP* livers, albeit the correct shape of the liver was not completely restored.

We recapitulated these results in adult male and female *TO(kras^{G12V})^{T/+}* zebrafish. In these experiments, total body mass was unaffected by single mutations in either *ranbp2* or *ahctf1*, or both mutant alleles together, independent of dox treatment (*Figure 7—figure supplements 2a and 3a*). However, when liver mass was expressed as a percentage of total body mass, this was reduced by 13% (*ranbp2*) and 28% (*ahctf1*). In adult males carrying both mutations, liver mass expressed as a percentage of total body mass was reduced by 54% (*Figure 7—figure supplement 2b and c*). Similar results were obtained in adult females (*Figure 7—figure supplement 3b, c*). Histological analysis of adult male livers demonstrated that hepatocyte integrity and organisation were improved (fewer abnormal nuclei and cytoplasmic vacuoles) in *ranbp2^{+/−};ahctf1^{+/−}* HETS (*Figure 7—figure supplement 4*). Thus, hyperplastic hepatocytes are highly susceptible to combinatorial targeting of NUP function, raising the question of whether this selective vulnerability could be exploited in the clinic.

## Combining Selinexor treatment with *ahctf1* heterozygosity blocks liver overgrowth in dox-treated *TO(kras^{G12V})^{T/+}* larvae

Selinexor is a selective inhibitor of nuclear export through its interaction with XPO1 (exportin). The drug has received FDA approval for the treatment of refractory or relapsed multiple myeloma and diffuse large B-cell lymphoma and is in clinical trials for several other cancers, including HCC (*Jans et al., 2019*; *Zheng et al., 2014a*), leading us to test the impact of Selinexor treatment in our larval model of HCC. We observed no reduction in liver volume in *2-CLiP* larvae exposed to 0.10–2.00 µM Selinexor from 5 to 7 dpf (*Figure 7—figure supplement 5a, b*). In contrast, *ahctf1^{+/+};TO(kras^{G12V})^{T/+}* larvae exhibited a dose-dependent reduction in liver volume when exposed to Selinexor and this was reduced further by *ahctf1* heterozygosity (*Figure 7—figure supplement 5c, d*). Indeed, at a 1.00 µM concentration of Selinexor, liver volume in *ahctf1^{+/−};TO(kras^{G12V})^{T/+}* larvae was indistinguishable from that of control *2-CLiP* larvae, indicating that liver overgrowth was completely blocked. These observations suggest that Selinexor could be an effective and selective suppressor of growth and proliferation in human HCC, and that its efficacy may be enhanced by combinatorial targeting with novel therapeutics designed to disrupt the function of the ELYS protein.

## Targeting NUPs is a promising therapeutic approach for HCC

To explore the clinical rationale for developing NUP inhibitors for the purpose of HCC treatment, we analysed gene expression data available for 366 patient samples in TCGA liver hepatocellular carcinoma (LIHC) dataset (*Figure 8—source data 1*) using the cBioPortal for Cancer Genomics (*Cerami et al., 2012*; *Gao et al., 2013*). Z-score values were allocated to mRNA expression levels (RNAseq V2 RSEM) for each of the genes encoding the 10 components of the NUP107–160 complex relative to the mean expression value of that gene in diploid samples. Using this approach, we found a higher percentage of HCC samples overexpressing NUP mRNAs (z-score >2) than those under-expressing NUP mRNAs (z-score <2). For example, in the case of *AHCTF1*, 32.4% of patient samples had a z-score of >2 relative to the mean expression of samples in the LIHC cohort that were diploid for *AHCTF1*. Conversely, only 0.82% of patient samples had a z-score of <−2 for *AHCTF1* relative to its expression in samples that were diploid for this gene (*Figure 8a*).

To determine whether differences in NUP gene expression were associated with overall survival, we analysed the clinical data available for patients in the LIHC dataset. Here, log-transformed mRNA expression z-scores were calculated relative to the expression distribution of the ten 107–160 NUP genes across all 366 patient samples. This was followed by a logrank test to compare the survival distributions of patients linked to samples in the high expression group (z-score >2) versus patients linked

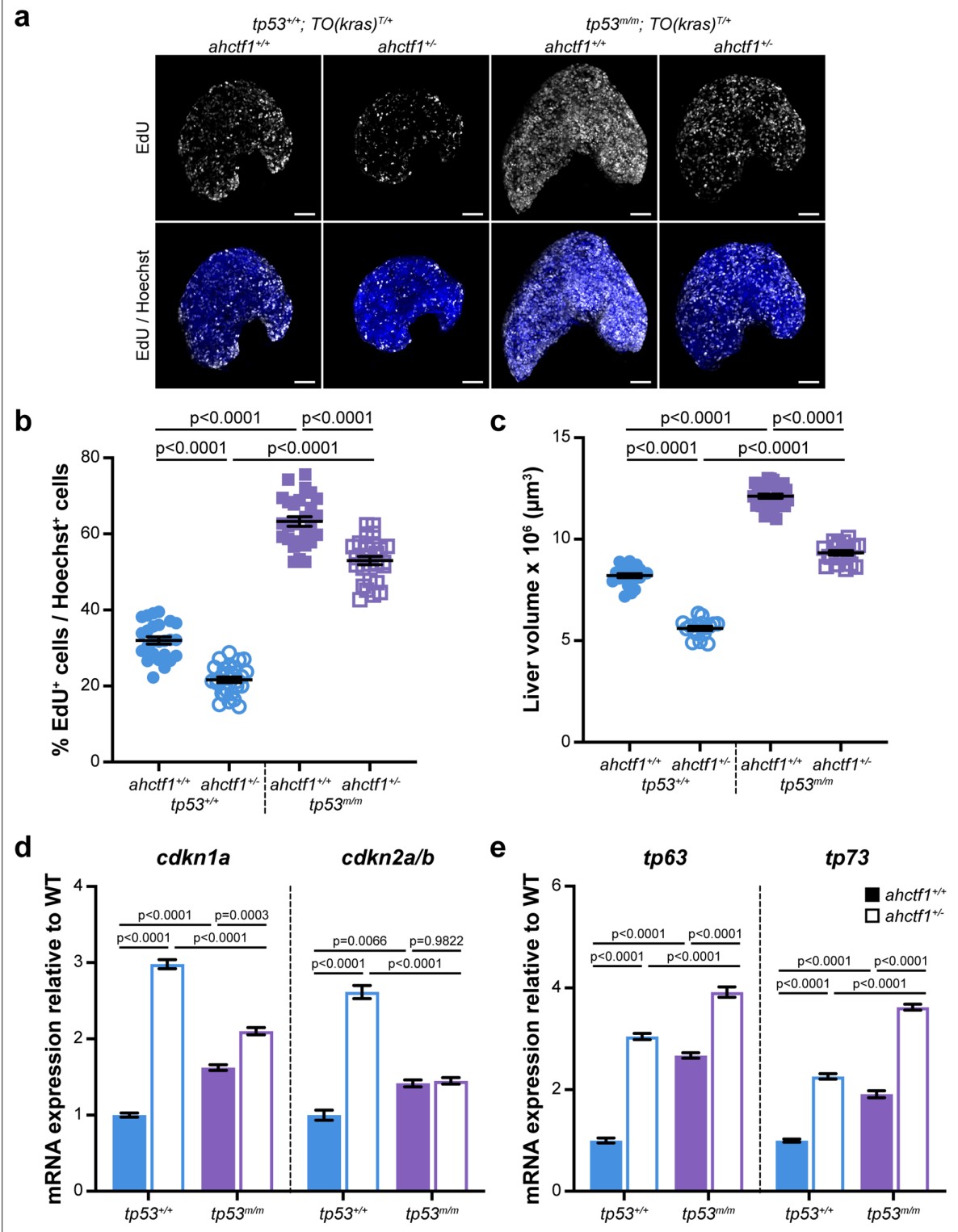

**Figure 6.** *ahctf1* heterozygosity restricts DNA replication in dox-treated *TO(kras^{G12V})^{T/+}* hepatocytes in the presence and absence of Tp53. (**a**) Representative maximum intensity projection images of EdU incorporation (white puncta) into *TO(kras^{G12V})^{T/+}* livers of the indicated *ahctf1* and *tp53* genotypes. Scale bar 25 μm. (**b**) Quantification of the percentage of EdU-positive nuclei per Hoechst 33342-positive nuclei ($n \geq 25$). (**c**) Impact of *ahctf1* heterozygosity and homozygous *tp53* mutation on liver volume in dox-treated *TO(kras^{G12V})^{T/+}* larvae ($n \geq 17$). (**d**) RT-quantitative PCR (RT-qPCR) analysis of mRNA expression of the cell cycle regulators, *cdkn1a* and *cdkn2a/b* and (**e**) *tp63* and *tp73* expression in *TO(kras^{G12V})^{T/+}* micro-dissected livers of the

*Figure 6 continued on next page*

*Figure 6 continued*

indicated *ahctf1* and *tp53* genotypes (*n* = 3 biological replicates). Data are expressed as mean ± standard error of the mean (SEM). Significance was assessed using a one-way analysis of variation (ANOVA) with Tukey's multiple comparisons test.

The online version of this article includes the following source data and figure supplement(s) for figure 6:

**Source data 1.** *Figure 6d*: *cdkna1* and cdkna2a/b mRNA expression analyses by RT-quantitative PCR (RT-qPCR) of micro-dissected livers from doxycycline (dox)-treated *TO(kras^{G12V})^{T/+}* zebrafish larvae; role of *ahctf1* and *tp53* mutation.

**Figure supplement 1.** *ahctf1* heterozygosity results in activation of a tp53 transcriptional programme in dox-treated adult male *TO(kras^{G12V})^{T/+}* zebrafish.

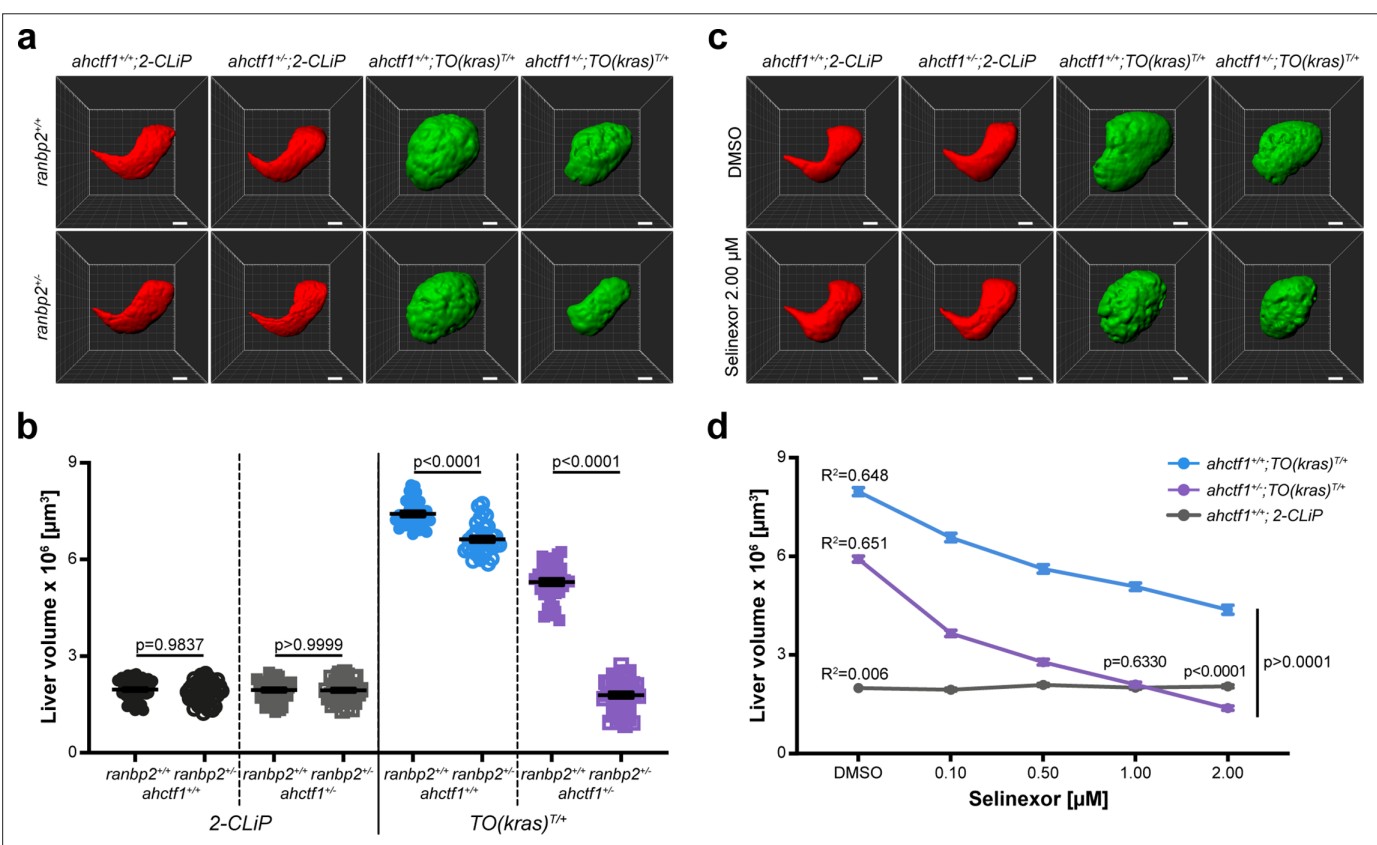

**Figure 7.** Combinatorial targeting of NUPs and Selinexor treatment restricts liver enlargement in dox-treated *TO(kras^{G12V})^{T/+}* hepatocytes. (**a**) Representative three-dimensional reconstructions of livers from *2-CLiP* and *TO(kras^{G12V})^{T/+}* larvae of the indicated *ahctf1* and *ranbp2* genotypes. Scale bar 25 µm. (**b**) Impact of *ahctf1* heterozygosity and *ranbp2* heterozygosity on liver volume in *2-CLiP* and *TO(kras^{G12V})^{T/+}* larvae (*n* ≥ 30). Significance was calculated using a one-way analysis of variation (ANOVA) with Tukey's multiple comparisons test. (**c**) Representative three-dimensional reconstructions of *TO(kras^{G12V})^{T/+}* livers of the indicated *ahctf1* genotype treated with dimethyl sulfoxide (DMSO) or 2.00 µM Selinexor from 5 to 7 dpf. (**d**) Dose-dependent impact of Selinexor treatment on liver volume in *2-CLiP*, *ahctf1^{+/+};TO(kras^{G12V})^{T/+}* and *ahctf1^{+/−};TO(kras^{G12V})^{T/+}* larvae (*n* ≥ 20). Data are expressed as mean ± standard error of the mean (SEM). Significance was calculated by linear regression analysis.

The online version of this article includes the following source data and figure supplement(s) for figure 7:

**Source data 1.** *Figure 7b*: Quantitation of liver volume (µm³) of *TO(kras^{G12V})^{T/+}* zebrafish larvae; role of *ahctf1* and *ranbp2* mutation.

**Figure supplement 1.** Genetic and morphological characterisation of the *ranbp2* mutant.

**Figure supplement 2.** Combinatorial targeting of *ahctf1* and *ranbp2* restricts *kras^{G12V}*-driven liver enlargement in adult male zebrafish.

**Figure supplement 3.** Combinatorial targeting of *ahctf1* and *ranbp2* restricts *kras^{G12V}*-driven liver enlargement in adult female zebrafish.

**Figure supplement 4.** Doxycycline treatment of adult male *TO(kras^{G12V})^{T/+}* zebrafish results in the development of histopathological features of hepatocellular carcinoma (HCC).

**Figure supplement 5.** Combination of Selinexor treatment and *ahctf1* heterozygosity reduces liver volume in *TO(kras^{G12V})^{T/+}* larvae but has no impact on liver volume in *2-CLiP* larvae.

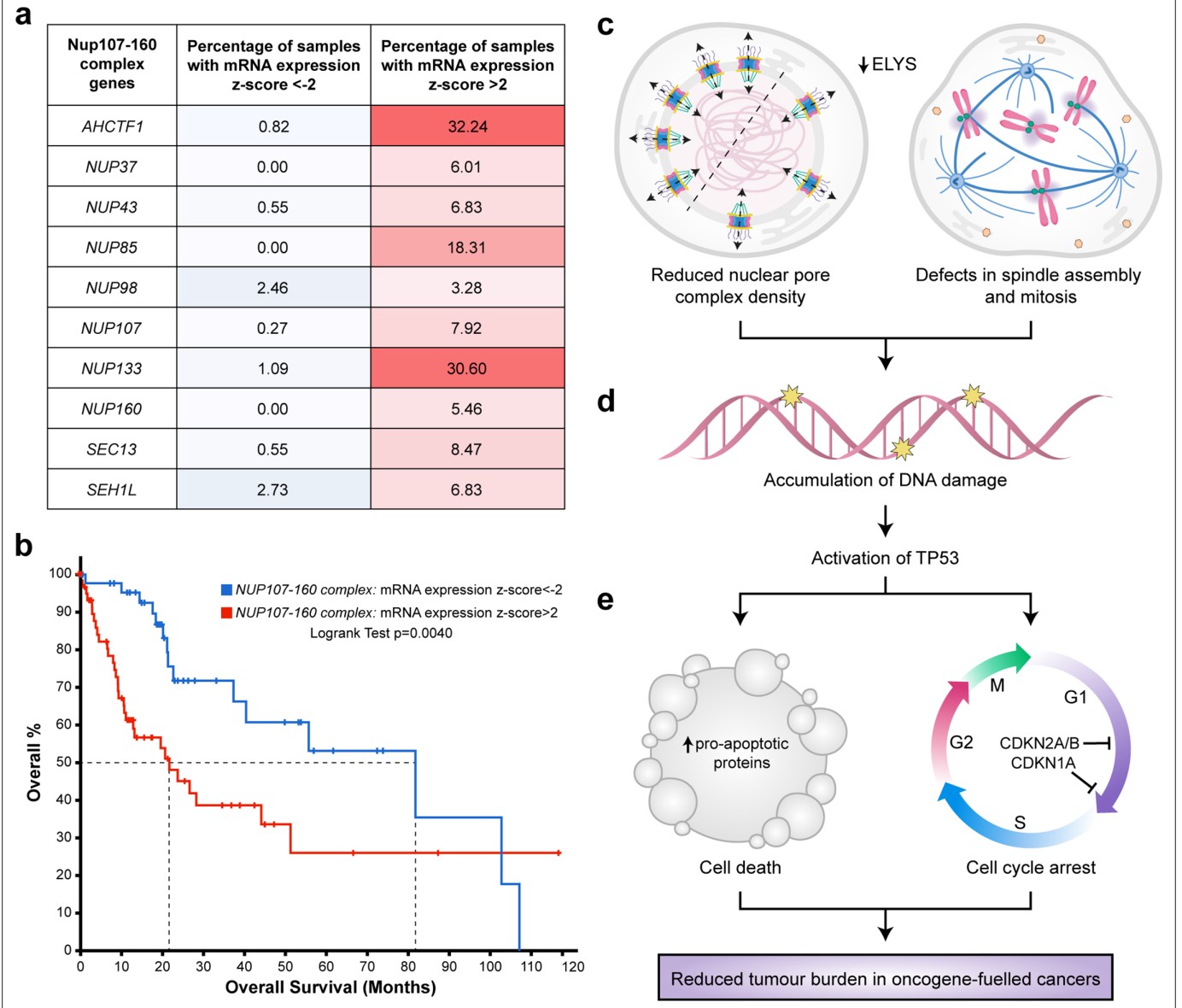

**Figure 8.** Targeting nuclear pore complexes (NPCs) is a promising therapeutic approach for hepatocellular carcinoma (HCC). (**a**) Heatmap based on percentage of hepatocellular carcinoma (LIHC) samples in The Cancer Genome Atlas (TCGA) with mRNA expression levels of NUP107–160 components that are under-expressed (z-score <−2: over 2 standard deviations below the mean expression of those genes in diploid samples) or overexpressed (z-score >2: over 2 standard deviations above the mean expression of those genes in diploid samples) analysed through the cBioPortal for Cancer Genomics (*Cerami et al., 2012*; *Gao et al., 2013*). See *Figure 8—source data 1*. (**b**) Probability of overall survival of HCC patients linked to samples with mRNA expression z-scores >2 for one or more NUP107–160 complex components (red line; 63 cases); median probability of overall survival 21.70 months. Probability of overall survival of HCC patients linked to samples with mRNA expression z-scores <−2 for one or more NUP107–160 complex components (blue line; 45 cases); median probability of overall survival 81.73 months. (**c**) Schematic depiction of two cellular processes disrupted by mild Elys depletion, leading to (**d**) accumulation of DNA damage and activation of Tp53 transcription programmes. (**e**) Induction of cell death and cell cycle arrest in hyperproliferative oncogene-expressing cancer cells.

The online version of this article includes the following source data and figure supplement(s) for figure 8:

**Source data 1.** *Figure 8a*: mRNA expression z-scores (RNAseq V2 RSEM) for *AHCTF1* and the nine other genes encoding components of the NUP107–160 complex, relative to the mean expression of each gene of interest in patient samples that are diploid for that gene.

**Figure supplement 1.** NUP107–160 complex gene expression in the The Cancer Genome Atlas (TCGA) liver hepatocellular carcinoma (LIHC) dataset is largely unaffected by unwanted variation.

to samples in the low expression (*z*-score <−2.0) group. We found that patients linked to samples with high mRNA expression *z*-scores (>2.0) for one or more NUP107–160 components appeared to do significantly worse than patients with samples in the low expression group (*z*-score <−2.0). Indeed, the median probability of overall survival for patients allocated to the high expression group was 21.70 months compared to 81.73 months (3.8-fold increase) for patients allocated to the low expression group (*Figure 8b*). We infer from these clinical data that patients with HCCs expressing high levels of mRNAs encoding ELYS and other components of the NUP107–160 complex may benefit most from strategies designed to target these NUP components therapeutically.

To examine the integrity of these interesting clinical data, we determined whether unwanted variation affected the expression levels of the NUP107–160 complex genes in samples of the LIHC dataset. If present, unwanted variation caused by factors such as differences in library size, tumour purity/heterogeneity and batch effects between participating laboratories, can significantly adversely affect downstream analyses including association between gene expression and survival outcomes (*Molania et al., 2022*). Since the 366 samples of the LIHC dataset were profiled across 19 sequencing plates that could potentially cause batch effects, we addressed this issue by performing analysis of variation (ANOVA) tests (*Figure 8—figure supplement 1*). These tests revealed that the expression of our genes of interest in the LIHC dataset was largely unaffected by unwanted variation, increasing confidence that the inverse relationship we report here between upregulated NUP107–160 gene expression and poorer overall survival rate is likely to be reliable.

## Discussion

In this paper we show that the presence of one mutant *ahctf1* allele markedly reduces hepatocyte growth and survival in a mutant *kras*-driven model of HCC. In our experiments *ahctf1* heterozygosity reduced the abundance of NPCs and introduced defects in mitotic spindle assembly and chromosome segregation only in hepatocytes expressing the *kras*$^{G12V}$ oncogene (*Figure 8c*). These perturbations combined to amplify the mild oncogenic stress exhibited by cells expressing the *kras*$^{G12V}$ oncogene, culminating in DNA damage, Tp53 activation (*Figure 8d*) and increased expression of Tp53 target genes that promote apoptosis and cell cycle arrest (*Figure 8e*). The exquisite vulnerability of hepatocytes co-expressing the *kras*$^{G12V}$ oncogene and a single mutated *ahctf1* allele demonstrates a synthetic lethal interaction between a powerful oncogene and heterozygous expression of an essential cellular gene. Such a finding is of significant interest considering the remarkable success of PARP1 inhibitors in the clinic and presents a potential new drug target for tumours expressing mutant KRAS, which has hitherto proven difficult to target directly.

Our results compliment a growing body of evidence that carcinogenesis places persistently high demands on essential cellular genes, including those encoding NUPs (*Sakuma et al., 2021*). Sakuma et al. used siRNAs targeting 28 individual NUP genes to reduce the abundance of NPCs in the human melanoma-derived cell line, A375, and showed that this caused cell death (*Sakuma et al., 2021*). Meanwhile, normal human pulmonary fibroblasts and retinal pigmental epithelium cells (RPE1), underwent a reversible cell cycle arrest instead, consistent with reports that quiescent cells generally require only low levels of NUP expression due to exceptional stability of assembled NPCs (*D'Angelo et al., 2009*; *Toyama et al., 2013*).

Mutations in *TP53*, or amplification/overexpression of its negative regulators *MDM2/MDM4*, occur in 30% of HCC cases (*Zucman-Rossi et al., 2015*). Therefore, it was pertinent to ask whether the potential therapeutic effect of inhibiting ELYS function in a clinical setting would be diminished in tumours not expressing WT TP53 protein. To address this question, we studied the impact of *ahctf1* heterozygosity in zebrafish that were homozygous for mutant *tp53*. We found that loss of Tp53 function permitted markedly higher rates of hepatocyte growth and liver enlargement in the larval, mutant *kras*-driven model of HCC, whether they were WT or heterozygous for *ahctf1*. However, total liver volume in *ahctf1* HETS was 35% less than in WT, meaning that the capacity of *ahctf1* heterozygosity to reduce liver volume was not entirely dependent on availability of WT Tp53. However, from a therapeutic perspective, we think that obtaining the full benefit of targeting ELYS protein in the clinic is likely to be affected by the mutational status of TP53.

Dysregulation of nucleocytoplasmic transport is a common feature in a broad spectrum of cancers, usually arising from altered expression of nuclear pore components and/or nuclear transport receptors (*Kau et al., 2004*). For example, the nuclear transport receptor XPO1 is frequently overexpressed

in cancers, leading to mis-localisation and inactivation of tumour suppressor proteins (*Zheng et al., 2014a*; *Hill et al., 2014*). We found that in our zebrafish HCC model, treatment with a clinically validated XPO1 inhibitor, Selinexor, produced a dose-dependent reduction in liver enlargement, which was strongly augmented by *ahctf1* heterozygosity. We also found that trans heterozygosity of *ahctf1* and *ranbp2* produced a synergistic reduction in liver volume in both *kras*$^{G12V}$-expressing larvae and adults. Normal livers not expressing the *TO(kras*$^{G12V}$*)* transgene were completely unaffected by Selinexor treatment or *ahctf1;ranbp2* trans heterozygosity, suggesting that new drugs targeting the ELYS protein and/or other NUPs could be combined effectively with drugs that target nucleocytoplasmic transport without forfeiting cancer selectivity.

To date, the molecular topology of the ELYS protein is poorly defined. Further high-resolution structural characterisation is required to advance our molecular understanding of the extent to which discrete domains within the ELYS protein contribute to its various interactions, and to reveal opportunities to inhibit such regions for the purpose of disabling ELYS function therapeutically. Crystal structures of ELYS have shown that the β-propeller and α-helical domains in the N-terminal half of the ELYS protein bind to NUP160, and cryo-electron microscopy revealed the association of the ELYS C-terminal region with nucleosomes (*Kobayashi et al., 2019*; *Bilokapic and Schwartz, 2013*). However, ELYS and other NUPs in the NUP107–160 complex generally lack features that would facilitate high affinity binding with small drug-like molecules (*Sakuma et al., 2021*; *Mitsopoulos et al., 2021*). Notwithstanding this, rapidly emerging technological advances in drug development may provide alternative strategies to target the ELYS protein. For example, carefully controlled proteasomal degradation of proteins that participate in the oncogenic process using proteolysis targeting chimeras (PROTACs) is a rapidly advancing field in cancer therapy (*Ocaña and Pandiella, 2020*). In parallel, emerging small molecule RNA-targeting technology is on course to permit direct modulation of the abundance of specific RNA transcripts for a variety of clinical purposes (*Mukherjee et al., 2020*). In summary, we believe the findings from our in vivo model of HCC provide a strong and feasible rationale for the development of novel therapeutics that target ELYS function as well as suggesting potential avenues for effective combinatorial treatments.

# Materials and methods

## Key resources table

| Reagent type (species) or resource | Designation | Source or reference | Identifiers | Additional information |
|---|---|---|---|---|
| Gene (*Danio rerio*) | *ahctf1* | Ensembl | ENSDARG00000077530 | |
| Gene (*Danio rerio*) | *ranbp2* | Ensembl | ENSDARG00000093125 | |
| Gene (*Danio rerio*) | *tp53* | Ensembl | ENSDARG00000035559 | |
| Genetic reagent Zebrafish (*Danio rerio*) | *ahctf1*$^{ti262}$ (TU) | European Zebrafish Resource Centre (EZRC) | ZDB-ALT-980203-1165 | Also known as *flotte lotte* |
| Genetic reagent Zebrafish (*Danio rerio*) | *ranbp2*$^{s452}$ (TU) | Heath Lab (see this paper; *Figure 7—figure supplement 1*) | | |
| Genetic reagent Zebrafish (*Danio rerio*) | *Tp53*$^{zdf1}$ (AB) | European Zebrafish Resource Centre (EZRC) | ZDB-ALT-050428-215201 | Also known as *Tp53*$^{M214K}$ |
| Genetic reagent Zebrafish (*Danio rerio*) | *Tg(fabp10:dsRed, ela31:GFP)*$^{gz12}$ | Zhiyuan Gong, National University of Singapore | PMID:18796162 | Also known as *Lipan* or 2CLiP |
| Genetic reagent Zebrafish (*Danio rerio*) | *Tg(fabp10:rtTA2s-M2; TRE2:EGFP-kras*$^{G12V}$*)*$^{gz32}$ | Zhiyuan Gong, National University of Singapore | PMID:23812423 | |
| Genetic reagent Zebrafish (*Danio rerio*) | *Tg(actb2:SEC-Hsa.ANXA5-mKate2,cryaa:mCherry)*$^{uq24rp}$ | Thomas Hall, Institute of Molecular Bioscience, University of Queensland | PMID:31754462 | |
| Antibody | mAb414 (Mouse monoclonal) | AbCam | Cat# ab24609 | IF (1:750) |
| Antibody | anti-γ-H2AX (Mouse monoclonal) | AbCam | Cat# ab11174 | IF (1:1000) |
| Antibody | anti-α-Tubulin DM1A (Mouse monoclonal) | Cell Signaling Technology | Cat#: 3873 | IF (1:1000) |

*Continued on next page*

*Continued*

| Reagent type (species) or resource | Designation | Source or reference | Identifiers | Additional information |
|---|---|---|---|---|
| Antibody | anti-cleaved Caspase3 (Rabbit polyclonal) | Cell Signaling Technology | Cat#: 9664 | IF (1:250) |
| Antibody | Donkey anti-Rabbit IgG (H+L) Highly Cross-Adsorbed Secondary Antibody, Alexa Fluor 647 (Donkey polyclonal) | Thermo Fisher Scientific | Cat#: A31573 | IF (1:500) |
| Antibody | Goat anti-Mouse IgG (H+L) Cross-Adsorbed Secondary Antibody, Alexa Fluor 647 (Goat polyclonal) | Thermo Fisher Scientific | Cat#: A21235 | IF (1:500) |
| Antibody | anti-Tp53 (9.1) (Mouse monoclonal) | AbCam | Cat# ab77813 | WB (1:500) |
| Antibody | anti-GAPDH (14C10) (Rabbit polyclonal) | Cell Signaling Technology | Cat#: 2118 | WB (1:1000) |
| Antibody | Goat Anti-Mouse Immunoglobulins/HRP (affinity isolated) (Goat polyclonal) | Dako | Cat#: P0447 | WB (1:5000) |
| Antibody | Goat Anti-Rabbit Immunoglobulins/HRP (affinity isolated) (Goat polyclonal) | Dako | Cat#: P0448 | WB (1:5000) |
| Commercial assay or kit | Active Ras Pull-Down and Detection kit | Thermo Fisher Scientific | Cat#: 16117 | |
| Commercial assay or kit | Click-iT Edu AF647 kit imaging kit | Invitrogen | Cat#: C10340 | |
| Chemical compound, drug | Selinexor (KPT-330) | Gift of Karyopharm Therapeutics | | XPO1 inhibitor |
| Chemical compound, drug | Doxycycline | Sigma Aldrich | Sigma D9891 | 20 µg/mL |

## Sequence-based reagents

### Table 1. Genotyping primers

| Name | Forward (5'–3') | Reverse (5'–3') |
|---|---|---|
| *ahctf1*$^{ti262}$ | TGACATGCATGCCCTCTCTG | TAGCTGCTCCTCGCTTACGT |
| *ranbp2*$^{s452}$ | CGCCGATCAAGAGGACGAAA | TGTCCGCCGTAACACTACTC |
| *tp53* wildtype | AGCTGCATGGGGGGGAT | GATAGCCTAGTGCGAGCACACTCTT |
| *tp53*$^{M214K}$ | AGCTGCATGGGGGGGAA | GATAGCCTAGTGCGAGCACACTCTT |

### Table 2. RT-qPCR primers

| Name | Forward (5'–3') | Reverse (5'–3') |
|---|---|---|
| *ahctf1*$^{ti262}$ | GGTGAGTCAGTGTGGGGAAC | CCAGCAGGGCATGAAGTGAT |
| *b2m* | GCGGTTGGGATTTACATGTTG | GCCTTCACCCCAGAGAAAGG |
| *bad* | AGCAGCACCTCACTGTTCCT | CCAGTTTCCAGCAAGTCCTC |
| *bax* | GGAGATGAGCTGGATGGAAA | GGGCCACTCTGATGAAGACA |
| *bbc3* | GATGCCTTCAGCTTGGAC | GCCTGGACACTTCCTGTTCT |
| *bcl2* | GGATCGAGGAAAATGGAGGT | AAAACGGGTGGAACACAGAG |
| *bclxl* | CAACCATATTCAACCCTGGA | TTCTTGCGATTTCCTGCT |

*Continued on next page*

*Continued*

| Name | Forward (5′–3′) | Reverse (5′–3′) |
|---|---|---|
| *bid* | ACCAGCGACCTACAGAGACC | TCTGCATTGACTGAAAGACCA |
| *bik* | TTGCTTCCACAGCTTCAAAA | ATGTAGTGCTGCGAGACCAG |
| *bim* | GCACTTTGATTTCCCTCAGC | TGGAGAAAGTCCGGTTCATC |
| *ccng1* | GTGCGGAGACGTTTTCCTT | AAGACAGATGCTTGGGCTGA |
| *cdkn1a* | CAAGCCAAGAAGCGTCTAGTG | AACGGTGTCGTCTCTGGTTC |
| *cdkn2a/b* | CGAGGATGAACTGACCACAGC | CAACAGCCAAAGGTGCGTTAC |
| *hrpt1* | GAGGAGCGTTGGATACAGA | CTCGTTGTAGTCAAGTGCAT |
| *mdm2* | TGACAAAGAAACTGGTAAGA | AAACATAACCTCCTTCATGGT |
| *pmaip1* | ATGGCGAAGAAAGAGCAAAC | TCATCGCTTCCCCTCCATTTG |
| *tbp* | CAGGCAACACACCACTTTAT | AAGTTTACGGTGGACACAAT |
| *tp53* | TCCACTCTCCCACCAACATC | GGGAACCTGAGCCTAAATCC |
| *tp63* | CGGCCTGTTTGGACTATTTC | ACTCCATGATGCCTTTCCAG |
| *tp73* | GGCCAATCCTCATCATCATC | TCCCTGAATGGTCTTCGTC |
| *Δ113tp53* | ATATCCTGGCGAACATTTGG | ACGTCCACCACCATTTGAAC |

## Zebrafish maintenance and strains

Zebrafish were maintained at 28°C on a 12 hr light/12 hr dark cycle. The mutant lines *ahctf1*[ti262] and *tp53*[M214K/M214K] (also known as *tp53*[e7/e7] and referred to herein as *tp53*[m/m]) have been described previously (*Chen et al., 1996*; *Berghmans et al., 2005*). The Tg(*fabp10:dsRed, ela3l:GFP*)[gz12] line, herein referred to as *2-CLiP*, expresses dsRed in the liver and GFP in the exocrine pancreas but carries no oncogenic transgenes or mutations (*Korzh et al., 2008*). The Tg(*fabp10:rtTA2s-M2;TRE2:EGFP-kras*[G12V])[gz32] line referred to as *TO(kras*[G12V])[T/+] and the cell death reporter line, Tg(*actb2:SEC-Hsa.ANXA5-mKate2,cryaa:mCherry*)[uq24rp] were described previously (*Chew et al., 2014*; *Hall et al., 2019*). The *ranbp2*[s452] mutant line was generated in the Liver[plus] ENU mutagenesis screen (*Ober et al., 2006*), and its genetic and morphological characterisation is presented for the first time in this paper.

## Inducing hepatocyte hyperplasia in larval zebrafish

To induce *kras*[G12V] expression, *TO(kras*[G12V])[T/+] embryos were treated with 20 μg/mL dox (Sigma, #D9891) at 2 dpf in E3 medium (5 mM NaCl, 0.17 mM KCl, 0.33 mM $CaCl_2$, 0.33 mM $MgSO_4$) with 0.003% 1-phenyl-2-thiourea (PTU; Sigma, #P7629) to suppress pigmentation. E3 medium was changed at 5 dpf and fresh dox (final conc. 20 μg/mL) added. For the Selinexor experiments, the drug (final concentration 0.10–2.00 μM) (Karyopharm) or 0.001% DMSO (vehicle control) was added at 5 dpf. Morphological and molecular analyses were performed at 7 dpf. To quantitate liver volume, larvae were anaesthetised with benzocaine (200 mg/L) and mounted in 1% agarose. Image acquisition was performed using an Olympus FVMPE-RS multiphoton microscope with a ×25 objective and Olympus FV30-SW software. Excitation wavelengths for GFP and RFP were 840 and 1100 nm, respectively. For volumetric analysis of whole livers, z-stacks with step size 2 μm were imported into ImageJ or Imaris software.

## Inducing HCC in adult zebrafish

To induce *kras*[G12V] expression, adult *TO(kras*[G12V])[T/+] zebrafish (>3 months post-fertilisation) were treated for 7 days with 20 mg/L doxycycline (Sigma, #D9891) with water and dox changed daily. Zebrafish were euthanised with benzocaine (1000 mg/L) and tumour burden quantified by liver-to-body mass ratio measurements.

## Genotyping

Genomic DNA (gDNA) was extracted from whole zebrafish larvae by incubation at 95°C in 50 µL of 50 mM sodium hydroxide (NaOH) for 20 min, followed by neutralisation with 5 µL of 1 M Tris–HCl (pH 8.0). Primer sequences for genotyping are listed in Table 1.

## mRNA expression analysis

Total RNA was extracted from independent pools of micro-dissected zebrafish livers using the RNeasy Micro Kit (QIAGEN, #74004). RNA integrity was assessed by a High Sensitivity RNA ScreenTape assay (Agilent, #5067-5579) on a 2200 TapeStation. cDNA was generated from 1 to 10 µg RNA using the Superscript III First Strand Synthesis System (Invitrogen, #18080051) and oligo(dT) priming according to the manufacturer's instructions. RT-qPCR was performed using a SensiMix SYBR kit (Bioline, #QT605-05) on an Applied Biosystems ViiATM7 Real-Time PCR machine. Expression data were normalised by reference to *hrpt1*, *b2m*, and *tbp*. LinRegPCR V11.0 was used for baseline correction, PCR efficiency calculation and transcript quantification analysis (*Ruijter et al., 2009*). Relative expression levels were calculated by the $2^{-\Delta\Delta Ct}$ method and all results were expressed as the mean ± standard error of the mean (SEM) of three independent pools of biological replicates. Primer sequences for RT-qPCR are listed in Table 2.

## Library preparation and RNA sequencing

RNA was isolated from individually micro-dissected livers and 100 ng per sample of RNA was used for RNA-sequencing library preparation using the TruSeq RNA Sample Prep Kit with Ribo-Zero depletion (Illumina). Indexed libraries were sequenced on a NextSeq 500 instrument (Illumina) to generate 80-bp paired end reads and yielding at least 20 million reads per sample. All samples were aligned to the GRCz11 build of the zebrafish genome using the align function from the Rsubread software package (v2.0.1) (*Liao et al., 2019*). In all cases at least 75% of fragments (read pairs) were successfully mapped. All fragments overlapping genes were counted using Rsubread's featureCounts function. Genes were identified using Ensembl annotation (v101). Differential expression analyses between the sample groups were then undertaken using the limma (v3.46.0) (*Ritchie et al., 2015*) and edgeR (v3.32.1) (*Robinson et al., 2010*) software packages. Expression-based filtering was performed using edgeR's filterByExpr function with default parameters. A total of 21,655 genes remained. Sample composition was normalised using the TMM method (*Robinson and Oshlack, 2010*). These data were transformed to log-counts per million (logCPM) with associated observation level precision weights using voom (*Law et al., 2014*). Linear models were fit to the data and robust empirical Bayes moderated *t*-statistics were utilised to identify differentially expressed genes (*Phipson et al., 2016*) through application of edgeR's voomLmFit pipeline. The false discovery rate was controlled below 5% using the Benjamini and Hochberg method. The mean difference plot was generated using limma's plotMD function and the heatmap using the pheatmap software package (v1.0.12). Gene set enrichment analysis was conducted with limma's cameraPR function with default parameters, and the associated barcode plots drawn with limma's barcodeplot function. Pathway analyses of the kyoto encyclopedia of genes and genomes (KEGG) database were accomplished using limma's kegga function. The data have been deposited in the GEO under accession number GSE220282.

## Western blot analysis

Pooled micro-dissected zebrafish livers were lysed in RIPA buffer (20 mM HEPES (4-(2-hydroxyethyl)-1-piperazineethanesulfonic acid), pH 7.9, 150 mM NaCl, 1 mM $MgCl_2$, 1% NP40, 10 mM NaF, 0.2 mM $Na_3VO_4$, 10 mM β-glycerol phosphate) supplemented with cOmplete Proteinase Inhibitor (Roche, #11836170001) and PhosTOP phosphatase inhibitors (Roche, #04906837001). Samples were incubated for 30 min on ice and the extracts cleared by centrifugation at 13,000 rpm for 20 min at 4°C. The protein concentration of samples was determined by BCA protein assay (Thermo Fisher Scientific, #23227). 25 µg of protein per lane were resolved on NuPAGE Novex Bis-Tris 4–12% polyacrylamide gels (Invitrogen, #NP0321BOX) and transferred onto nitrocellulose blotting membranes (Amersham Protran, #10600003). Membranes were blocked with 5% bovine serum albumin (BSA) in phosphate-buffered saline (PBS) for 1 hr at room temperature (RT) and incubated with primary antibodies 1:500 Anti-p53 (9.1) mouse mAb (Abcam, #ab77813) overnight at 4°C and 1:1000 Anti-GAPDH (14C10) Rabbit mAb (Cell Signalling Technology, #2118) for 1 hr at RT. Secondary antibodies: Goat anti-mouse

HRP (Dako, #P0447) and Goat anti-Rabbit HRP (Dako, #P0448) were used at 1:5000 and incubated with membranes for 1 hr at RT. Signals were developed using Amersham ECL Western Blotting Detection Kit (Cytiva, # RPN2108) and imaged on a Chemidoc Touch (Bio-Rad). Relative protein quantitation was calculated based on normalised integrated intensity.

## Active Ras pull-down and detection

Total lysate was isolated from 7 dpf zebrafish larvae and equal amounts of protein (50 μg) were used as input for the pull-down of activated GTP-Ras proteins using the Active Ras Pull-Down and Detection kit (Thermo Fisher, #16117) according to the manufacturers' instructions.

## Cell death analysis

To assess apoptosis, 7 dpf *TO(kras$^{G12V}$);Annexin 5-mKate* zebrafish larvae were fixed in 4% paraformaldehyde (PFA) overnight at 4°C and livers isolated by micro-dissection. Image acquisition was performed using a Zeiss LSM 880 microscope with a ×20 objective and ZEN software. Excitation wavelengths for mKate and GFP were 560 and 900 nm, respectively. Liver volume was quantified and 3D segmentation of Annexin 5-mKate signals were performed in ImageJ.

## Analysis of DNA replication in S-phase

EdU incorporation was used to assess the percentage of cells in S-phase of the cell cycle. Briefly, live zebrafish larvae (7 dpf) were incubated at 28°C in 2 mM EdU (Invitrogen, #C10340) in E3 medium for 2 hr followed by a further incubation in fresh E3 medium for 1 hr. Larvae were euthanised using benzocaine (1000 mg/L; Sigma, #PHR1158) prior to removal of the liver by micro-dissection. EdU labelling was carried out using the Click-iT Edu Alexa Fluor 647 (AF647) imaging kit (Invitrogen, #C10340), according to the manufacturer's instructions. The livers were co-stained with Hoechst 33342 (Thermo Fisher, #62249). Image acquisition was performed using an Olympus FVMPE-RS multiphoton microscope with excitation wavelengths of 950 nm for Hoechst 33342 dye and 1160 nm for AF647 (Thermo Fisher, #A21235). The number of Hoechst 33342- and EdU-positive cells was quantified using Arivis Vision4D software.

## Nuclear pore analysis

Larvae fixed in 4% PFA overnight at 4°C were embedded in 4% low-melting temperature agarose and transverse sections collected at 200 μm intervals using a vibrating microtome (Leica VT 1000S). Sections were blocked with 1% BSA in PBS/0.3% Triton X-100 and incubated with a 1:750 dilution of mAb414 (Abcam, #ab24609) at 4°C overnight. Sections were then incubated with 1:500 anti-mouse AF647 and Hoechst 33342 at room temperature for 1 hr. Sections were mounted and imaged using a Zeiss LSM880 Fast Airyscan Confocal microscope with a ×63 objective. 3D segmentation of cells was performed in ImageJ using EGFP signal for *TO(kras$^{G12V}$)$^{T/+}$* larvae or F-actin stained with 1:200 rhodamine phalloidin for *TO(kras$^{G12V}$)$^{+/+}$* larvae. An outline was drawn around the nuclear periphery and segmented using the Hoechst signal. mAb414 fluorescence intensity was calculated for the nuclear periphery and for the cytoplasm. NPC density was calculated by finding maxima for mAb414 at the nuclear surface of 5 nuclei per liver.

## Cryosectioning and immunofluorescence microscopy analysis

Dissected livers were fixed in 4% PFA overnight at 4°C and washed with PBS/0.1% Tween 20 before incubation in 30% sucrose in PBS overnight at 4°C. Livers were aligned in a tissue mould, embedded in OCT and frozen on dry ice. The livers were sectioned at 10μm intervals using a Thermo Fisher Scientific Microm HM550 cryostat. Sections were washed with PBS before blocking with 10% FCS in PBS/0.3% Triton X-100. Incubation with primary antibodies was performed at 4°C overnight, while incubation with secondary antibodies was performed at room temperature for 1 hr. Antibodies used in this work were: 1:2000 a-Tubulin DM1A (CST, #3873), 1:1000 γ-H2AX (gift of James Amatruda), 1:250 cleaved caspase-3 (CST, #9664), 1:500 anti-rabbit AF647 (Thermo Fisher Scientific, #A31573) and 1:500 anti-mouse AF647 (Thermo Fisher, #A21235). Prolong Diamond Antifade reagent with DAPI (Thermo Fisher #P36962) was used for slide mounting. A Zeiss LSM880 Fast Airyscan Confocal microscope with a ×63 objective was used for image acquisition and ImageJ for image analysis.

## TCGA LIHC dataset analysis

Gene expression values for patient samples in the LIHC dataset in the TCGA, PanCancer Atlas were analysed using cBioPortal (*Cerami et al., 2012*; *Gao et al., 2013*). The NUP107–160 complex genes *AHCTF1*, *NUP37*, *NUP43*, *NUP85*, *NUP98*, *NUP107*, *NUP133*, *NUP160*, *SEC12*, and *SEH1L* were queried in samples with mRNA data (RNA Seq V2 RSEM). Patient samples were assigned to groups with high or low expression of one or more NUP107–160 complex genes based on $z$-scores $>2$ and $<-2.0$, respectively, relative to diploid samples. Patient samples that overlapped the selected groups were excluded from the overall survival analysis.

## Statistical analysis

Data are expressed as mean ± standard error of the mean (SEM), unless indicated otherwise, and the number of biological replicates indicating samples from individual animals/livers, or pools of individual animals/livers for each experiment are stated in the figure legends. p values were calculated using Student's *t*-tests (two-tailed, followed by Welch's correction) when comparing two groups, and by ANOVA followed by Tukey's post hoc test when comparing multiple groups. Chi-square tests were used to analyse the distribution of mitotic cells and the number of mitotic cells exhibiting an aberrant phenotype. The effect of Selinexor treatment on liver volume was analysed by linear regression, regressing liver volume against Selinexor concentration. All analysis was performed using GraphPad Prism V7.03 (GraphPad software) and p values $\leq 0.05$ were considered statistically significant.

## Acknowledgements

The authors thank Tyson Blanch, Cameron Mackey, Elizabeth Grgacic, and Bryan Ko for excellent zebrafish husbandry and Ellen Tsui for histology. We are grateful to Assoc Prof Kimberley Evason for reviewing the zebrafish liver histology, Prof James Amatruda for the gift of rabbit polyclonal antibody to zebrafish γ-H2AX and Karyopharm Therapeutics for the gift of Selinexor. We thank Drs Andrew Cox, Leigh Coultas, Alexandra Garnham, Quentin Gouil, Assoc Prof Brendon Monahan, and Prof Gordon Smyth for bioinformatics support and insightful discussions.

## Additional information

### Competing interests

Elke A Ober: Reviewing editor, *eLife*. Didier YR Stainier: Senior editor, *eLife*. The other authors declare that no competing interests exist.

### Funding

| Funder | Grant reference number | Author |
| --- | --- | --- |
| National Health and Medical Research Council | Project GNT 1024878 | Joan K Heath |
| Australian Government | Graduate Student Training Program | Kimberly J Morgan |
| Ludwig Institute for Cancer Research | Ludwig Member Support Package | Joan K Heath |

The funders had no role in study design, data collection, and interpretation, or the decision to submit the work for publication.

### Author contributions

Kimberly J Morgan, Conceptualization, Data curation, Formal analysis, Investigation, Methodology, Writing – original draft, Writing – review and editing; Karen Doggett, Conceptualization, Formal analysis, Supervision, Investigation, Methodology, Writing – original draft, Writing – review and editing; Fansuo Geng, Formal analysis, Investigation; Stephen Mieruszynski, Formal analysis; Lachlan Whitehead, Kelly A Smith, Benjamin M Hogan, Cas Simons, Gregory J Baillie, Ramyar Molania, Anthony T Papenfuss, Thomas E Hall, Formal analysis, Investigation, Methodology; Elke A Ober, Didier YR

Stainier, Zhiyuan Gong, Methodology; Joan K Heath, Conceptualization, Formal analysis, Supervision, Funding acquisition, Investigation, Methodology, Writing – original draft, Project administration, Writing – review and editing

### Author ORCIDs
Kimberly J Morgan http://orcid.org/0000-0001-7871-2583
Karen Doggett http://orcid.org/0000-0002-8676-8461
Benjamin M Hogan http://orcid.org/0000-0002-0651-7065
Thomas E Hall http://orcid.org/0000-0002-7718-7614
Didier YR Stainier http://orcid.org/0000-0002-0382-0026
Joan K Heath http://orcid.org/0000-0001-6955-232X

### Ethics
All husbandry and experimental procedures performed on zebrafish followed standard operating procedures and were conducted with the approval of the Animal Ethics Committees of the Walter and Eliza Hall Institute and The University of Melbourne, Parkville, Victoria, Australia. WEHI-AEC approved project 2019.014, project title: Zebrafish disease models and mechanisms.

### Decision letter and Author response
Decision letter https://doi.org/10.7554/eLife.73407.sa1
Author response https://doi.org/10.7554/eLife.73407.sa2

## Additional files

### Supplementary files
• Transparent reporting form

### Data availability
A new RNA sequencing dataset has been deposited in GEO under accession ID GSE220282. Existing datasets analysed during the current study are available in the cBioPortal Cancer Genomics database (http://www.cbioportal.org). All data generated/analysed during this study are included in the Figures and figure supplements and Source Data files are provided for Figures 1–8.

The following dataset was generated:

| Author(s) | Year | Dataset title | Dataset URL | Database and Identifier |
|---|---|---|---|---|
| Garnham A, Heath J, Doggett K, Morgan K | 2022 | Molecular characterisation of a mutant kras-driven zebrafish model of hepatocellular carcinoma | https://www.ncbi.nlm.nih.gov/geo/query/acc.cgi?acc=GSE220282 | NCBI Gene Expression Omnibus, GSE220282 |

The following previously published dataset was used:

| Author(s) | Year | Dataset title | Dataset URL | Database and Identifier |
|---|---|---|---|---|
| TCGA PanCancer Atlas | 2018 | Liver Hepatocellular Carcinoma | https://www.cbioportal.org/study/summary?id=lihc_tcga_pan_can_atlas_2018 | cBioPortal Cancer Genomics database, Genomics |

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
