## [Editor Report]

We believe that the study demonstrates the importance of nuclear pore complex components for Kras/p53 driven liver tumors. The findings made here in zebrafish may stimulate additional preclinical and mechanistic studies to test the role of nuclear pore components in cancer.

---

## [Decision Letter]

**Decision letter after peer review:**

Thank you for submitting your article "Elys deficiency constrains Kras-driven tumour burden by amplifying oncogenic stress" for consideration by *eLife*. Your article has been reviewed by 3 peer reviewers, including Hao Zhu as the Reviewing Editor and Reviewer #1, and the evaluation has been overseen by Richard White as the Senior Editor.

Essential revisions:

1. Please determine if tp63 and tp73 are compensating for tp53 loss. One possibility is to knockdown these genes (perhaps using CRISPR/Cas9) in the presence and absence of tp53. The finding that tp63/tp73 knockdown suppresses apoptosis in tp53 mutants but not in tp53 wild-type zebrafish would support your stated hypothesis.

2. Please determine if ahctf1 heterozygosity indeed decreases liver tumor burden in the TO(kras) model. This experiment in adult zebrafish should be fairly straightforward. The finding that ahctf1+/- have decreased liver tumor burden (quantified by liver-to-body-mass ratio and histologic analysis) would bolster the significance of this work greatly.

3. When referring to larval liver size, the language of the paper should be changed to "liver overgrowth" rather than "tumor burden".

4. Statistical analysis (error bars) are missing on Figure 6c and d.

5. In the krasG12V model, it would be helpful to distinguish the contribution of increased cell death vs decreased cell proliferation to the change in liver size seen with heterozygous ahctf1. Is this predominantly due to decreased proliferation?

6. Please determine if the heterozygous ahctf1 state blunts the overall level of Ras activity in krasG12V animals.

7. It would be helpful to see detailed transcriptional profiling of hepatocytes in the krasG12V model with the heterozygous ahctf1 mutation, and to assess the effects of Selinexor. GSEA type analysis offers a way to start untangling the effects of these pathways. Moreover this analysis could provide insight on the relevance of this model to human HCC. Please let un know if it is feasible to generate this data within a reasonable time frame.

8. Functions of Achtf1 in regard to chromatin regulation could be compromised in this model. Scholz et al. (Nat Gen 2019) report that Ahctf1 is involved in increasing Myc expression via gene gating mechanism. What are effects on Myc in this system?

*Reviewer #1 (Recommendations for the authors):*

I see that this dependency in mammalian cells is described in the discussion, but I think that adding some mammalian cancer cell line data showing the effect of one allele ahctf1 deletion would be highly informative. Also, if the authors could cite a paper showing that het mice are viable and normal, that would be useful.

*Reviewer #2 (Recommendations for the authors):*

1. To determine if tp63 and tp73 are compensating for tp53 loss, it would be interesting to knock down these genes (perhaps using CRISPR/Cas9) in the presence and absence of tp53. The finding that tp63/tp73 knockdown suppresses apoptosis in tp53 mutants but not in tp53 wild-type zebrafish would support your hypothesis.

2. It would be REALLY interesting to see if ahctf1 heterozygosity indeed decreases liver tumor burden in the TO(kras) model. This experiment in adult zebrafish should be fairly straightforward. The finding that ahctf1+/- have decreased liver tumor burden (quantified by liver-to-body-mass ratio and histologic analysis) would bolster the significance of this work greatly.

3. When referring to larval liver size, the language of the paper should be changed to "liver overgrowth" rather than "tumor burden".

---

## [Author Response]

Essential revisions:1. Please determine if tp63 and tp73 are compensating for tp53 loss. One possibility is to knockdown these genes (perhaps using CRISPR/Cas9) in the presence and absence of tp53. The finding that tp63/tp73 knockdown suppresses apoptosis in tp53 mutants but not in tp53 wild-type zebrafish would support your stated hypothesis.

Our claim that "the beneficial effect of *ahctf1* heterozygosity to reduce tumour burden persists in the absence of functional Tp53 due to compensatory increases in the levels of *tp63* and *tp73*” was an overstatement and has now been deleted. While our data demonstrated higher levels of *tp63* and *tp73* mRNA expression in response to *ahctf1* heterozygosity and Tp53 loss, we presented no data to indicate that this was due to activation of a compensation mechanism. Indeed, the mechanism(s) underlying genetic compensation are not well understood and therefore difficult to test (El-Brolosy MA and Stainier DYR 2017. PLoS Genet 13, e1006780).

While Reviewer 2 suggests that Crispr/Cas9 knock-out experiments may help us to resolve this question, we think that the results of these experiments would be of limited relevance in a clinical context, since the human *TP63* and *TP73* genes are rarely mutated in cancer. Also, we would need to generate viable triple mutant zebrafish to undertake these experiments, and these will be extremely rare/non-viable, as is the case in mice (Van Nostrand JL *et al.* Cell Death Diff doi:10.1038/cdd.2016.128). Instead, we have replaced the contentious text in a new paragraph (see below), which can be found in the revised manuscript on page 14, lines 291-298.

“We also noted that mRNAs encoding the Tp53 family members, Tp63 and Tp73 were upregulated by 3.0-fold and 2.3-fold, respectively, in heterozygous *ahctf1* livers, compared to WT *ahctf1* livers, in the presence of WT Tp53 (blue bars; Figure 6e). Interestingly, in the absence of WT Tp53 (purple bars, Figure 6e), *ahctf1* heterozygosity further enhanced the expression of *tp63* and *tp73*, albeit mildly. Since *tp63* and *tp73* are not direct Tp53 target genes, this raises the possibility that *tp63* and/or *tp73* expression mediate a Tp53-independent role in the inhibition of hepatocyte growth in response to *ahctf1* heterozygosity. Tp63 and Tp73 have several direct target genes in common with Tp53, including the pro-apoptotic genes, *bbc3*, *pmaip1*, *bim* and *bax*, and the cell cycle arrest gene, *cdkna1*.”

2. Please determine if ahctf1 heterozygosity indeed decreases liver tumor burden in the TO(kras) model. This experiment in adult zebrafish should be fairly straightforward. The finding that ahctf1+/- have decreased liver tumor burden (quantified by liver-to-body-mass ratio and histologic analysis) would bolster the significance of this work greatly.

We thank Reviewer 2 for suggesting these experiments, and we agree that the new data we obtained from adults has greatly improved the significance of our study. In these experiments, adult zebrafish were treated daily with doxycycline for 7 d to induce *kras^G12V^* expression, resulting in a 9-fold increase in the liver-to-body mass ratio in males (new Figure 1e), with similar results in females (Figure 1e). When we introduced *ahctf1* heterozygosity into this setting we observed a 30% reduction in liver-to-body mass ratio in both male and female adults (Figure 1e). These new results with adult zebrafish essentially recapitulate the findings we made with larvae and highlight the concordance between the developmental and adult models. We also carried out RNA sequencing and gene set enrichment analysis to compare differential gene expression between WT and dox-treated TO(*kras^G12V^)^T/+^* larval livers (Figure 1—figure supplement 1). The significant correlation between the differential gene expression signature of KrasG12V expressing hepatocytes, compared to wildtype hepatocytes, and the differential gene signature of 369 patient samples in the Liver hepatocellular (LIHC) dataset of The Cancer Genome Atlas (TCGA), compared to healthy livers, provide additional confidence that our larval model of HCC is relevant to human HCC.

3. When referring to larval liver size, the language of the paper should be changed to "liver overgrowth" rather than "tumor burden".

In all cases, including the title of the revised manuscript, where we used the term “tumour burden”, we have replaced the term with liver enlargement or liver overgrowth.

4. Statistical analysis (error bars) are missing on Figure 6c and d.

We have completed the statistical analysis as requested on what is now Figure 3 panels c, d (previously Figure 6c and d).

5. In the krasG12V model, it would be helpful to distinguish the contribution of increased cell death vs decreased cell proliferation to the change in liver size seen with heterozygous ahctf1. Is this predominantly due to decreased proliferation?

We think this question is difficult to address, because the relative contributions of the two processes may vary with time. Our data show definitively that by 7 dpf, the impact of *ahctf1* heterozygous mutation has disrupted multiple cellular processes, leading to a 40% increase in the number of hepatocytes expressing Annexin 5 (dying cells), and a 40% decrease in the number of hepatocytes incorporating EdU over a 2 h incubation (fewer cells in S-phase). Both responses are likely to contribute to the reduction in liver volume observed in response to *ahctf1* heterozygosity. It is worth stating that in our experiments, we captured snapshots of apoptosis and DNA replication in the livers of larvae at 7 days post-fertilisation after 5d of dox treatment/KrasG12V expression. To answer the Reviewer’s question properly, we would need to monitor the behaviour of individual cells over time. If such experiments were technically possible, we think that some cells that undergo growth arrest in response to dox treatment might ultimately succumb to apoptosis (unless dox treatment is withdrawn) while other cells might enter into a state of prolonged senescence. However, given the technical challenges, we did not attempt to test this in the current manuscript.

6. Please determine if the heterozygous ahctf1 state blunts the overall level of Ras activity in krasG12V animals.

We have addressed this interesting question thoroughly in new Figure 1g, h. To do this, we used a commercial RAS-RBD pulldown kit followed by western blot analysis to determine the levels of activated GTP-bound Kras protein. Our results demonstrate that the levels of GTP-bound Kras protein, expressed as a proportion of total Kras protein, do not change in response to *ahctf1* heterozygosity. We conclude from these data that the potentially therapeutic value of reduced *ahctf1* expression in a cancer setting is not caused by inhibiting Kras activity.

7. It would be helpful to see detailed transcriptional profiling of hepatocytes in the krasG12V model with the heterozygous ahctf1 mutation, and to assess the effects of Selinexor. GSEA type analysis offers a way to start untangling the effects of these pathways. Moreover this analysis could provide insight on the relevance of this model to human HCC. Please let un know if it is feasible to generate this data within a reasonable time frame.

Please also see our response to comment #2. We used RNAseq to address the relevance of our larval model to human HCC. Specifically, we performed differential gene expression analysis to identify up- and downregulated genes in cohorts of *ahctf1*^+/+^ (WT) larvae versus dox-treated *ahctf1*^+/+^(WT);*kras^G12V^* larvae. We used gene set enrichment analysis to compare these differentially regulated transcripts with the gene expression signature of 369 patient samples in the Liver hepatocellular carcinoma (LIHC) dataset versus healthy liver samples in the TCGA. These analyses revealed a significant association between the patterns of gene expression in our larval model of zebrafish HCC and those of human HCC (Figure 1—figure supplement 1c, d).

The genetic experiments we report in Figures 4, 5, 6 show that WT Tp53 is required for the reductions in liver enlargement (Figure 4), apoptosis (Figure 5) and DNA replication (Figure 6) that occurs in response to *ahctf1* heterozygosity in dox-treated *kras^G12V^* larvae. We also used RT-qPCR to show that a Tp53-mediated transcriptional program was activated in these *ahctf1* heterozygous livers (Figure 5). Similarly, in **adult** livers, *ahctf1* heterozygosity triggered the upregulation of Tp53 target genes, including pro-apoptotic genes (*pmaip1*, *bbc3, bim* and *bax*) and cell cycle arrest genes (*cdkn1a* and *ccng1*) (new Figure 6—figure supplement 1). These results show that to obtain the full potential of *ahctf1* heterozygosity in reducing growth and survival of *Kras^G12V^*-expressing hyperplastic hepatocytes requires activation of WT Tp53. This is an important conclusion from our paper that is likely to be relevant in a clinical setting, for instance in patient selection, if ELYS inhibitors are developed for the treatment of HCC in which the KRAS/MAPK pathway is activated.

Also, one reviewer mentions performing genome-wide transcriptional profiling of hepatocytes in the krasG12V model in response to *ahctf1* heterozygosity and the presence and absence of Selinexor treatment. While these are potentially interesting experiments, they are substantial in nature and not crucial for the main messages of our paper. Therefore, we respectively contend that they are beyond the scope of the current manuscript.

8. Functions of Achtf1 in regard to chromatin regulation could be compromised in this model. Scholz et al. (Nat Gen 2019) report that Ahctf1 is involved in increasing Myc expression via gene gating mechanism. What are effects on Myc in this system?

The Scholz, 2019 and Gondor, 2022 papers from the same group, are very interesting in that they demonstrate a novel role for the ELYS protein in addition to the ones we pursued in our paper. The authors showed that in HCT116 cells, a human colorectal cancer cell line in which proliferation is driven by aberrant WNT/CTNNB1 signalling, the longevity of nascent MYC mRNA was increased by accelerating its movement from the nucleus to the cytoplasm, thereby preventing its degradation by nuclear surveillance mechanisms. The authors showed that siRNA knockdown of AHCTF1 in HCT-116 cells reduced the rate of nuclear export of MYC transcripts without changing the transcriptional rate of the MYC gene. They proposed a mechanism that depended on the formation of a complex chromatin architecture comprising transcriptionally active MYC and CCAT1 alleles plus proteins including β-Catenin, CTCF and ELYS. Together these interacting components guided nascent MYC mRNA molecules to nuclear pores, enhanced their export to the cytoplasm to be translated, resulting in activation of a MYC transcriptional program that induced expression of pro-proliferation genes. In theory, this role of ELYS in protecting MYC from nuclear degradation might extrapolate to other cancer settings where MYC expression is elevated. While interplay between MYC and mutant KRAS to enhance cancer growth has been previously reported, to date, most emphasis on this interaction has focused on the role of mutant KRAS in increasing the stability of the MYC protein, for example via RAS effector protein kinases (ERK1/2 and ERK5) that stabilise MYC by phosphorylation at S62 (Farrell and Sears, 2014: https://doi.org/10.1101/cshperspect.a014365) (Vaseva and Blake 2018: DOI:https://doi.org/10.1016/j.ccell.2018.10.001). While we appreciate the novelty of the recent papers, the current findings are limited to β-Catenin activated HCT-116 cells and may not be relevant to our zebrafish model of mutant Kras-driven HCC. Accordingly, we have not allocated a high priority to following this up in our current manuscript.

Reviewer #1 (Recommendations for the authors):I see that this dependency in mammalian cells is described in the discussion, but I think that adding some mammalian cancer cell line data showing the effect of one allele ahctf1 deletion would be highly informative.

Yes, we cited the Sakuma *et al.* paper in the Discussion (page 18, second paragraph) as an example of a cell line-based study demonstrating NUP dependence in cancer cells versus normal cells. In a separate study, a genome-wide CRISPR/Cas9 targeting/gene essentiality screen (Meyers et al., 2017) showed that *AHCTF1* is essential for the viability of all 342 cancer cell lines examined (curated in the GenomeCRISPR database http://genomecrispr.dkfz.de/#!/) (Rauscher et al., 2017). While neither of these approaches address the specific question posed by this reviewer, they are consistent with the notion of NUP inhibitors offering a viable therapeutic window for cancer treatment. Against this background, we are hesitant to embark on the vagaries of picking one or more human cancer cell lines in which to delete one allele of *AHCTF1* as this could turn out to be a time-consuming and frustrating exercise. We prefer to stand by our in vivo approach, which we believe offered a cleaner, well-controlled system for our study.

Also, if the authors could cite a paper showing that het mice are viable and normal, that would be useful.

We have cited the paper on page 6, line 115 (Okita *et al.* ref 8), which shows that *Ahctf1* het mice are viable and normal.

Reviewer #2 (Recommendations for the authors):2. It would be REALLY interesting to see if ahctf1 heterozygosity indeed decreases liver tumor burden in the TO(kras) model. This experiment in adult zebrafish should be fairly straightforward. The finding that ahctf1+/- have decreased liver tumor burden (quantified by liver-to-body-mass ratio and histologic analysis) would bolster the significance of this work greatly.

We agree with these comments and accordingly, we performed several new experiments in adult fish. We have addressed this comprehensively under item #2

3. When referring to larval liver size, the language of the paper should be changed to "liver overgrowth" rather than "tumor burden".

Notwithstanding our demonstration that the gene expression profile exhibited by our larval model of HCC is significantly associated with that of human HCC, we have used the terms liver enlargement and liver overgrowth instead of tumour burden throughout the manuscript, including the title (addressed under item #3).